# A Recent Advance in the Diagnosis, Treatment, and Vaccine Development for Human Schistosomiasis

**DOI:** 10.3390/tropicalmed9100243

**Published:** 2024-10-15

**Authors:** Tanushri Chatterji, Namrata Khanna, Saad Alghamdi, Tanya Bhagat, Nishant Gupta, Mohammad Othman Alkurbi, Manodeep Sen, Saeed Mardy Alghamdi, Ghazi A. Bamagous, Dipak Kumar Sahoo, Ashish Patel, Pankaj Kumar, Virendra Kumar Yadav

**Affiliations:** 1Department of Biosciences, Institute of Management Studies Ghaziabad (University Courses Campus), Adhyatmik Nagar, NH-09, Ghaziabad 201015, Uttar Pradesh, India; tanyabhagat.mscbt2022@imsuc.ac.in; 2Department of Biochemistry, M A Rangoonwala College of Dental Sciences and Research Centre, 2390-B, K.B. Hidayatullah Road, Azam Campus, Camp, Pune 411001, Maharashtra, India; namratakhanna6@gmail.com; 3Department of Clinical Laboratory Sciences, Faculty of Applied Medical Sciences, Umm Al-Qura University, Makkah 21955, Saudi Arabia; ssalghamdi@uqu.edu.sa (S.A.); mokurbi@uqu.edu.sa (M.O.A.); 4Engineering Department, River Engineering Pvt Ltd., Toy City, Ecotech–III, Greater Noida 201306, Uttar Pradesh, India; nishant.gupta@riverengg.com; 5Department of Microbiology, Dr. Ram Manohar Lohia Institute of Medical Sciences, Vibhuti Khand, Gomti Nagar, Lucknow 226010, Uttar Pradesh, India; sen_manodeep6@yahoo.com; 6Respiratory Care Program, Clinical Technology Department, Faculty of Applied Medical Science, Umm Al-Qura University, Makkah 21955, Saudi Arabia; smghamdi@uqu.edu.sa; 7Department of Pharmacology and Toxicology, Faculty of Medicine, Umm Al-Qura University, Makkah 21955, Saudi Arabia; gabamagous@uqu.edu.sa; 8Department of Veterinary Clinical Sciences, College of Veterinary Medicine, Iowa State University, Ames, IA 50011, USA; dsahoo@iastate.edu; 9Department of Life Sciences, Hemchandracharya North Gujarat University, Patan 384265, Gujarat, India; adpatel@ngu.ac.in; 10Department of Environmental Science, Parul Institute of Applied Sciences, Parul University, Vadodara 391760, Gujarat, India; pankaj.kumar25135@paruluniversity.ac.in; 11Marwadi University Research Center, Department of Microbiology, Faculty of Sciences, Marwadi University, Rajkot 360003, Gujarat, India

**Keywords:** schistosomiasis, bilharzia, praziquantel, urogenital, tropical neglected disease

## Abstract

Schistosomiasis, which affects a large number of people worldwide, is among the most overlooked parasitic diseases. The disease is mainly prevalent in sub-Saharan Africa, southeast Asian countries, and South America due to the lack of adequate sanitation. The disease is mainly associated with poor hygiene, sanitation, and contaminated water, so it is also known as a disease of poverty. Three *Schistosoma* species (*S. mansoni*, *S. japonicum*, and *S. haematobium*) cause significant human infections. Co-infections with *Schistosoma* and other parasites are widely common. All these parasites may cause intestinal or urogenital schistosomiasis, where the disease may be categorized into the acute, sensitized, and chronic phases. The disease is more prevalent among school children, which may cause anemia and reduce development. Chronic infections frequently cause significant liver, intestinal, and bladder damage. Women exposed to contaminated water while performing normal duties like washing clothes might acquire urogenital schistosomiasis (UGS), which can cause tissue damage and raise the risk of blood-borne disease transmission, including human immunodeficiency virus (HIV) transmission. Praziquantel (PZQ) is the World Health Organization (WHO)-prescribed treatment for individuals who are known to be infected, but it does not prevent further re-infections with larval worms. Vaccine development and new molecular-based diagnosis techniques have promised to be a reliable approach to the diagnosis and prevention of schistosomiasis. The current review emphasizes the recent advancement in the diagnosis of schistosomiasis by molecular techniques and the treatment of schistosomiasis by combined and alternative regimes of drugs. Moreover, this review has also focused on the recent outbreak of schistosomiasis, the development of vaccines, and their clinical trials.

## 1. Introduction

Schistosomiasis is a snail-borne parasitic infection caused by trematode flatworms of the genus *Schistosoma* [1]. It is also referred to as bilharziasis/snail fever/Katayama fever, a poverty-based illness that, if untreated, leads to life-threatening pathologies [2]. It is one of the most overlooked tropical parasitic infections caused by blood-residing flukes, which has the highest economic impact in tropical regions, second only to malaria. Due to its life-threatening nature and its prevalence in the tropical parts of the world, it has a high mortality rate [3]. Every year, nearly 4400 to 200,000 individuals die due to different types of schistosomiases in the world, whereas in 2021 alone, 11,792 victims died due to schistosomiasis [4]. The largest number of schistosomiases have indeed been documented in tropical parts of Asia, Africa, and South America [5]. The World Health Organization (WHO) has set global goals for schistosomiasis as a public health problem to prevent and eliminate schistosomiasis by 2020 and 2025, respectively. As per the latest WHO report, schistosomiasis is endemic in 78 countries, out of which 51 countries have moderate-to-high transmission that requires preventive treatment [6]. Moreover, it also claims that, globally in 2021, schistosomiasis affected around 251.4 million individuals who are in requirement of preventive therapy for schistosomiasis, out of which 75.3 million patients have been cured. Moreover, WHO reports also reveal that during 2021, due to COVID-19, there was decreased provision of neglected tropical disease interventions and treatment coverage for bilharzia. Comparing data from 2019 and 2020 (COVID-2019 pandemic), during the COVID-19 pandemic, there was a 27% decrease in treatment coverage. This poses a serious threat to the progress and achievements made in previous decades [6].

The WHO has categorized schistosomiasis into three classes based on the level of endemicity and the need for preventive chemotherapy (PCT). These classes are namely as follows: (i) non-endemic—areas where schistosomiasis is not present; (ii) endemic but no PC needed—areas having schistosomiasis but PC is not required; and (iii) endemic, PC required—areas where schistosomiasis is present and PC is needed [7].

Acute schistosomiasis (Katayama fever) can develop weeks or months (approximately 2–8 weeks) post the primary infection as a systemic outcome of schistosomula, as they move through the circulation from the lungs to the liver, as well as to egg antigens [8]. A free-living larva of a freshwater lake parasite penetrates the skin of swimmers and persons who use infected lake or river water to wash clothes, obtain water, etc. Such an infection in the case of swimmers is called swimmers’ itch or cercarial dermatitis. This disease is related to Katayama fever and is more prevalent in those who have an initial phase of schistosomal infection. In persistently infected populations, *Schistosoma japonicum* and other species can lead to acute schistosomiasis, which can progress to infections with enhanced severity [9].

Earlier, several investigators reported on the epidemiology and challenges of schistosomiasis, including urogenital schistosomiasis (UGS), by focusing on either an individual country or the world. UGS is primarily caused by *Schistosoma haematobium*, which is a significant parasitic disease affecting millions globally, particularly in sub-Saharan Africa (SSA). It manifests with a range of symptoms and complications, including hematuria, bladder cancer [10], and increased susceptibility to human immunodeficiency virus (HIV) [11]. The WHO reviewed and changed the description of urinary schistosomiasis to UGS because of the genital problems resulting from schistosomiasis, which can be mistakenly diagnosed as a sexually transmitted infection or cervical cancer [12,13,14]. The inflammatory responses to the presence of eggs produced by adult worms are responsible for infection-associated pathologies. The eggs produce proteolytic enzymes, which help in their migration to the intestine and bladder for shedding. When eggs become trapped in cells or embolize into the spleen, liver, lungs, or brain, the enzymes trigger an eosinophilic inflammatory response. Long-term symptoms may vary depending on the schistosome species as mature worms of various species move to different locations, whereas many infections cause minor symptoms (anemia and malnutrition), which are frequent in endemic locations [7].

Recently, a cross-sectional investigation demonstrated a high prevalence of schistosomiasis in the adults of Madagascar, particularly in rural areas. The prevalence and risk factors of schistosomal infection were assessed by using a semi-quantitative polymerase chain reaction (PCR) from samples received from 1482 adult volunteers. The prevalence of *Schistosoma mansoni* was highest in the Andina region at 59.5%, but *S. haematobium* infection was most prevalent in the Ankazomborona region at 61.3%. Furthermore, the coinfection rate of both species in the Madagascar region was approximately 3.3%. According to the findings, the percentage of infection was higher in males (52.4%) and those who were important contributors to the incomes of their families (68.1%). On the other hand, those whose occupation was not farming and those who were older were found to be protective factors for disease. It was concluded that there is a requirement to expand schistosomiasis control programs to include adults, who are often excluded from PCT programs [15].

In another study, investigators highlighted the recent prevalence of UGS in African countries. Further, the investigators revealed the various challenges being faced for the diagnosis of this disease in field visits. Finally, the investigators suggested the application of rapid molecular assays as a valuable method for field diagnosis of UGS [16].

In the present review, the authors have emphasized schistosomiasis, including their life cycle, different species, epidemiology, clinical manifestations, recent outbreaks, available and emerging diagnoses, and treatment methods. Here, investigators have emphasized the various types of *Schistosoma* species and their worldwide distribution. Furthermore, the authors have also emphasized the emerging drugs and vaccine and their clinical trials for schistosomiasis. Here, the objective was to provide a recent advancement in the diagnosis, treatment, prevention control, outbreak, and vaccine development in the endemic areas of *Schistosoma*.

## 2. Schistosome Life Cycle

The life cycle of the schistosome, parasites responsible for schistosomiasis, is complex and involves multiple stages across different hosts, with freshwater snails as intermediate hosts and mammals, including humans, as definitive hosts. Figure 1a shows the schematic infection cycle of *schistosomiasis* in human hosts, while Figure 1b shows the life cycle of *S. japonicum* and intestinal schistosomiasis, and Figure 1c depicts the life cycle of *S. haematobium* and resulting genital schistosomiasis. The life cycle of the parasite is characterized by a series of morphological and biochemical changes that enable the parasite to adapt to various environments and hosts. This lifecycle is crucial for understanding the transmission dynamics and potential control strategies for schistosomiasis, a major public health concern in many tropical regions.

The lifecycle of the parasite can be divided into several stages: the egg stage, miracidium stage, sporocyst stage, cercaria stage, schistosomulum stage, and adult stage. During the egg stage, schistosome eggs are released into the environment through the feces or urine of the definitive host, typically humans or other mammals. These eggs hatch in freshwater, releasing miracidia, which are free-swimming larval forms. In the miracidium stage, the miracidia must find and penetrate a suitable intermediate host, usually a freshwater snail. This stage is critical as it initiates the asexual phase of the life cycle [17]. The sporocyst stage takes place in the snail, where the miracidia transform into sporocysts, where they undergo asexual reproduction to produce cercariae. This stage is marked by significant gene expression changes to adapt to the snail host. In the cercaria stage, cercariae are released from the snail into the water. Cercariae are free-swimming and must find a definitive host to penetrate the skin and enter the bloodstream. This stage is crucial for transmission to the mammalian host [18]. The schistosomulum stage takes place in the definitive host, where the cercaria transforms into a schistosomula. This stage involves migration through the host’s tissues, where they mature into adult worms. The schistosomula must evade the host’s immune system and adapt to the host’s internal environment. In the final stage, i.e., the adult stage, adult schistosomes reside in the blood vessels of the definitive host, where they mate and produce eggs. The adult worms exhibit sexual dimorphism, with distinct roles for males and females in reproduction [19]. The lifecycle is completed when the eggs are excreted by the host, continuing the cycle [20].

There are various types of adaptations/interactions between the host and parasite during their life cycle. One is intermediate host adaptation where the schistosomes have evolved to exploit the physiological and environmental conditions of their snail hosts, optimizing cercaria production.

Another one is definitive host adaptation in the mammalian host, where the schistosomes must navigate immune responses and physiological barriers. They manipulate host stress responses and immune evasion strategies to ensure survival and reproduction [21,22].

## 3. Signs, Symptoms, and Pathophysiology of Schistosomiasis

### 3.1. Signs and Symptoms of Schistosomiasis

Schistosomiasis can be categorized as intestinal schistosomiasis (ItS) and UGS. ItS is characterized by intestinal damage, hypertension in the abdominal blood vessels, and liver enlargement, among others. The UGS not only affects the urinary tract rather than being a gynecological disease in women, but it also affects the urinary bladder, ureter, and kidney. Furthermore, *S. haematobium* may also be associated with the development of bladder cancer [23]. It is possible to treat pathological conditions of schistosomiasis, but preventive measures are required to incorporate and focus on the high-risk population, which could be implemented through proper management plans.

The clinical signs of schistosomiasis were presented by McManus et al., 2020, where acute schistosomiasis is mainly present with a feverish syndrome, while chronic schistosomiasis arises due to long-standing illness from living in poor rural places. Sometimes, the host tissues may have trapped schistosome eggs, which may cause immunopathological reactions that may further cause inflammatory and obstructive illness in the urinary system (*S. haematobium*) or intestine-related illness, hepatosplenomegaly, and hepatic fibrosis (*S. mansoni* and *S. japonicum*). So, these parasites may cause immunopathogenic manifestations from schistosomiasis due to the immunology and host–parasite interaction in schistosomiasis [24].

Acute schistosomiasis generally begins to appear weeks to months after infection due to the development of the worm, the generation of eggs, the secretion of egg antigen, as well as the immunological complex and florid granulomatous reactions of the host. Katayama syndrome, the acute form of infection, manifests with fever, myalgia malaise, eosinophilia, headache, drowsiness, and stomach discomfort that lasts 2–10 weeks. *S. japonicum*, *S. mansoni*, and *S. mekongi* are mainly accountable for chronic intestine-related illness. With the progress of the infection, there is a granulomatous response, and the eggs are downregulated through multiple mechanisms, which is accompanied by intermittent stomach discomfort, nausea, and gastrointestinal bleeding. The frequency of the recurrence of symptoms depends on the severity of the infection [25]. Furthermore, there may be localized gastrointestinal characteristics, like isolated mucosal hyperplasia, pseudo-polyposis, and polyposis interleaved with regular bowel function (appendix). Periportal (Symmer’s pipe-stem) fibrosis preserves hepatic function, making this different from cirrhosis and liver disorders. 

When there is a sign of ascites and hematemesis by esophageal varices as a side effect of portal hypertension, a deadly variant of the disease can be seen. Granulomatous pulmonary arteritis may lead individuals with advanced hepatic fibrosis to experience significant pulmonary hypertension. It typically takes 5 to 15 years from the first infection to severe fibrosis. Nevertheless, children as young as 6 years old can develop periportal fibrosis, requiring early screening and treatment [7].

Female genital schistosomiasis (FGS) is a significant yet often overlooked condition caused by the parasitic infection of *S. haematobium*, which primarily affects the female reproductive tract [26,27]. This condition is prevalent in regions where schistosomiasis is endemic and can lead to a range of gynecological and reproductive health issues [28]. For instance, there will be the formation of inflammatory lesions in the reproductive organs, oviducts, cervix, vaginal, and vulva when vesical plexus eggs move to the genital system. There will be the presence of white sand-like patches in the lower genital tract, which are connected to neovascularization and friable mucosa [29,30]. Women with FGS may present with a range of symptoms, including pelvic pain, stress incontinence, vaginal bleeding, and genital disfigurement. These symptoms can mimic other gynecological conditions, making diagnosis challenging [31]. In some cases, there could be infertility and an increased risk of abortion. In due course of time, due to the damage to the genital tract, there may be increased indications of infection. This may lead to an increase in the chance of transmission of HIV infection, which may lower the chances of successful treatment of *S. haematobium*-related FGS [32].

In males, UGS may be accompanied by hematospermia, orchitis, prostatitis, dyspareunia, and oligospermia [33]. The conditions are quite curable and responsive to treatment regimens in comparison to FGS [34,35]. Figure 2A depicts a cystoscopic view of the posterior bladder mucosa, while Figure 2B shows a histological section from a bladder mass biopsy exhibiting *S. haematobium* ova.

*S. japonium* may also be associated with colorectal cancer, which investigations carried out by several investigators evidence. Chronic inflammation is considered to be one of the basic causes behind the induction of colorectal cancer [33,36].


Figure 2(**A**) The presented visual depicts a cystoscopic field of the posterior bladder mucosa exhibiting UGS lesions, granulomas, ulcers, and tumors. (**B**) Histology from the bladder mass biopsy shows *S. haematobium* ova (black circles) and Squamous cell carcinoma (SCC) of the bladder (expression of sialyl Lea). The figure is adapted with permission from Vale et al. (2015) and Santos et al. (2021) [37,38].
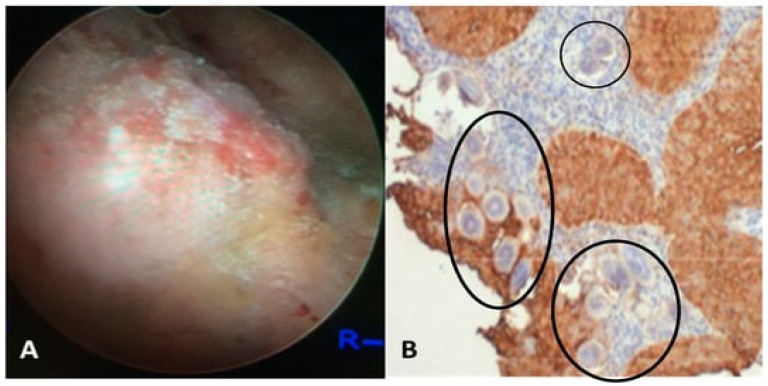



Figure 3 shows the metabolites present in urine with urogenital schistosomiasis and UGS-linked SCC and are absent from the urine samples of healthy individuals. Here, the investigators have divided the metabolites extracted from the U ring into three categories: catechol estrogen-like, catechol-estrogen-DNA adducts, and derivatives of 8-oxodG. The second metabolite might have formed due to the interaction of derivatives of catechol estrogen and host DNA. In contrast, the third metabolite might have formed due to the release of nitrogenous bases from DNA and/or its oxidation [37].

There are several endemic regions of schistosomiasis, especially in rural areas. The population often recognizes the signs and symptoms of the disease, such as hematuria and difficulty in urination. These symptoms are frequently identified in the vernacular and may be ascribed to prevalent endemic disorders. Nevertheless, some individuals establish the link between these signs and the underlying cause: snails in freshwater ecosystems that harbor the parasite. The lack of awareness about the role of snails in the transmission cycle contributes to the spread of the disease. People continue to use contaminated water sources without recognizing the health risks involved. Public health education is essential for narrowing this knowledge gap to enhance the prevention of schistosomiasis.

### 3.2. Pathophysiology of Schistosomiasis

The pathophysiology of schistosomiasis could be summarized as follows: Firstly, the parasite larva enters the skin and migrates to the lungs, where it molts and enters pulmonary circulation. Once it reaches the pulmonary system, it could reach the liver, leading to its maturation and pairing up. Further, the adult worms then migrate to their desired site of egg deposition, depending on the types of species. Generally, *S. mansoni* and *S. japonicum* prefer the intestinal mesenteric veins, while *S. haematobium* prefers the vesical and pelvic veins of the urinary tract. Once the female worms lay eggs, the sharp spines help them transverse the blood vessel wall of the host and ultimately reach the intestine or bladder’s lumen. The excretion of eggs with feces or urine marks the completion of the life cycle of *Schistosoma*. During this period, some of the eggs cannot reach the lumen and get trapped in the tissues, thereby inducing a granulomatous reaction, which is the major cause of pathology in schistosomiasis. The formation of a granuloma consists of an agglomeration of white blood cells (macrophages, lymphocytes, eosinophils, and fibroblasts) that surround and isolate the eggs and release cytokines and chemokines, collagen, and other matrix proteins. The cytokines and chemokines are responsible for the modulation of the immune response and inflammation, while collagen and matrix proteins are associated with fibrosis and the scarring of the affected organs. Together, these features are responsible for numerous complications which depend on the involvement of the organ. Generally, in ItS, the granulomas lead to ulceration, bleeding, obstruction, and malabsorption of the intestine, while fibrosis may cause portal hypertension, splenomegaly, ascites, and variceal bleeding of the liver.

UGS leads to significant pathophysiological changes in the host body, particularly affecting the urinary tract. Among patients, it causes a range of health issues, from mild symptoms to severe complications, including bladder cancer. The pathophysiology of UGS involves complex interactions between the parasite and the host’s immune system, leading to chronic inflammation and tissue damage. The urinary tract pathology involves bladder and ureteral damage, where the infection is characterized by the deposition of eggs in the bladder and ureters. This leads to granulomatous inflammation and fibrosis [32]. Fibrosis could lead to hydronephrosis, renal failure, bladder cancer, and infertility [37]. This can result in bladder wall thickening, vesical polyps, and obstructive uropathy, which are common findings in infected individuals [41]. This is followed by hydronephrosis and hydroureter. These conditions are frequently observed as complications of urinary schistosomiasis, resulting from the obstruction of urine flow due to egg-induced lesions in the urinary tract. This is followed by proteinuria and hematuria, where there are elevated levels of urine albumin and micro-hematuria. Both signs are indicative of urinary tract pathology and are used as markers for assessing the severity of the disease. 

The pathology of FGS involves complex interactions between the parasite and the immune system of the host. This results in several clinical manifestations and potential complications like egg granuloma formation, mucosal lesions and vascular changes, and the involvement of the upper reproductive tract. FGS is characterized by the presence of *S. haematobium* eggs in the female genital tract, leading to the formation of granulomas. These chronic lesions are frequently located in the vulva, vagina, and cervix and can cause symptoms such as pelvic pain, vaginal discharge, and irregular menstruation [30,42]. The presence of schistosome eggs can lead to the development of abnormal mucosal blood vessels, including dilated and tortuous venules in the cervicovaginal mucosa. These changes are indicative of a persistent tissue reaction to the eggs and can contribute to the pathology of FGS [30]. FGS can also affect the upper reproductive tract, leading to complications such as ectopic pregnancy and infertility. The involvement of the fallopian tubes and peritoneum is not uncommon in endemic areas and can result in conditions like hydrosalpinx and chronic salpingitis [42]. The diagnosis of FGS often requires a high index of suspicion, especially in women with a history of travel to or residence in endemic areas. Gynecological investigation may reveal friable cervical lesions, and the presence of schistosome eggs can be confirmed through cytology smears or biopsies. Eosinophil cationic protein levels in vaginal lavage extracts have been investigated as a potential diagnostic marker for FGS. Elevated ECP levels indicate an inflammatory immune response, although the test’s sensitivity and specificity are limited [37].

The immune response to schistosomiasis is complex and variable, and it depends on the phases of the disease, the load of parasites, genetic background, and environmental factors. These immune responses could be categorized into three phases, which are described in Figure 4: sensitization, acute, and chronic [7]. 

The disease progresses from an active phase, characterized by significant egg excretion, to an inactive phase with reduced egg output. The intensity of infection correlates with the severity of uropathy and mortality risk. Infection intensity and the associated pathologies show age-related patterns, with a peak in younger individuals and a decline in older age groups, possibly due to the development of partial immunity [43]. 

The host’s immune response to *S. haematobium* involves a Th2-type immune reaction, which can lead to chronic inflammation and tissue damage. This immune response is further modulated in cases of schistosomiasis-induced bladder cancer. Variations in the genetic makeup of the parasite may influence the severity of the disease, with certain genotypes being associated with more severe lesions. Chronic infection with *S. haematobium* is a known risk factor for bladder cancer. The molecular mechanisms involve oxidative stress and immune modulation, which contribute to tumorigenesis [44].

## 4. Distribution of *Schistosoma* Species and Schistosomiasis

The three most common ectoparasites of schistosome species that infect human hosts *are S. haematobium*, *S. mansoni*, and *S. japonicum*. *S. mansoni* infections have been mainly reported from some parts of South American countries, along with sporadic reports in the Arabian Peninsula. Africa and pockets of the Middle East have the main prevalence of *S. haematobium* and *S. mansoni*, while *S. japonicum* is only prevalent in Asian countries like China, the Philippines, and Sulawesi. In SSA, both *S. haematobium* and *S. mansoni* can be prevalent simultaneously. Other species include *Schistosoma mekongi*, which is found in the Mekong River region, and *Schistosoma guineensis*, which is prevalent in central and western Africa along *Schistosoma intercalatum* [45]. The habitat of *Schistosoma* is widely distributed because of the existence of varied water bodies worldwide. Hence, numerous schistosomal species are spread globally, with a particular range of snail hosts. The distribution according to varied geographical regions is briefed in Table 1, and country-wise [46], the distribution of some species of this parasite is shown in Figure 5.


tropicalmed-09-00243-t001_Table 1Table 1*Schistosoma* species and their geographical distribution.Schistosoma SpeciesGeographical DistributionReferences
*S. haematobium*
Middle Eastern Africa[9]
*S. japonicum*
Southeast Asia, Eastern Asia[47]
*S. mansoni*
Southern America, Caribbean islands, Africa[48]*Schistosoma margrebowiei*, *S. guineensis*, *S. intercalatum*Middle East, Africa[49]
*S. mekongi*
Southeast Asia, Asia[50]



Figure 5Global distribution of schistosomal infection adapted with permission from [51].
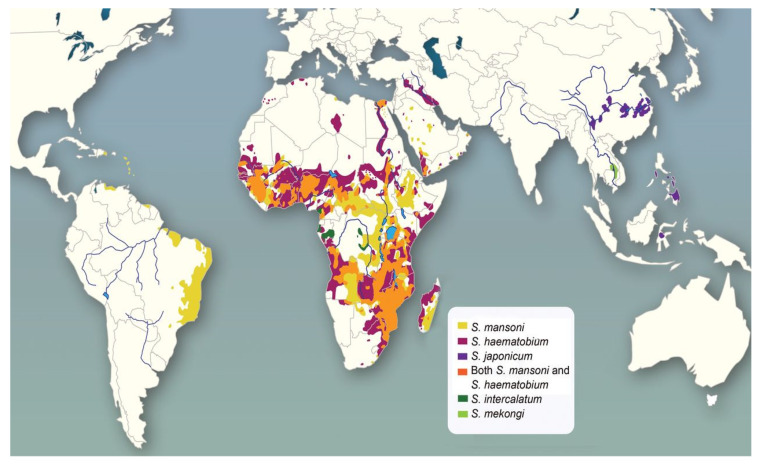



Schistosomiasis affects all age groups. Individuals in frequent contact with water, like fishermen, laundrymen, etc., are usually affected the most. The chronic form of schistosomiasis occurs from recurrent exposure to infectious cercariae [7]. In pediatric cases, the initial phase of infection is usually observed at 2 years, with the intensity of the disease rising gradually during the next 10 years as the latest worm colonies continue to grow within the child’s body. The frequency of infection increases remarkably among infants and young children [7]. The prevalence and intensity are highest in young adolescents (10–14 years of age), which eventually decreases as the child reaches adulthood, i.e., at 15–19 years of age. According to serosurveys, it was concluded that in endemic regions, almost all residents living there for a long time get infected with schistosomes at some point in their life. However, in regions with the typical transmission, 20–40% of adults and about 60- 80% of school-age children can remain actively infected [52].

Lai et al. [53] carried out a study on the spatial distribution of schistosomiasis and therapy requirements in SSA. It was a geostatistical investigation focusing on the countries of Africa. A survey of *S. haematobium* and *S. mansoni* was carried out in 9318 and 9140 unique locations, respectively. A reduction in the number of cases was noticed from 2000 onwards, yet estimates suggest that 163 million (95% Bayesian credible interval (CrI), 155 million to 172 million; 18.5%, 17.6–19.5) people in the sub-Saharan African population were infected in 2012. The maximum occurrence of schistosomiasis was among school-aged children (SAC) (52.8%; 95% credible interval, 48.7–57.8) in Mozambique in 2015. An outbreak with a prevalence of less than 10% was also observed among SAC in low-risk countries like Burundi, Equatorial Guinea, Eritrea, and Rwanda. Annually, about 123 million (95% CrI, 121–125 million) doses of praziquantel (PZQ) are needed for SAC, and 247 million doses are needed for the whole population (239–256 million) [53].

## 5. Morbidities and Co-Morbidities of Schistosomiasis

The progression of the *Schistosoma* infection causes systemic morbidities, which include anemia (blood loss, leading to iron insufficiency). Hepatic hormonal hepcidin mediates the internal iron rerouting that results in anemia. The secretion of interleukin 6 (IL-6), a pro-inflammatory cytokine, is triggered by infection, causing this release [54]. In addition, the malnutrition and impaired growth of children are observed in infected subjects. As it is accompanied by continuous inflammation, normal development, iron metabolic cycle, physical fitness, and mental faculties are all affected. These problems make therapy difficult, and the window of opportunity for effective preventative care is quite small. 

The etiology of *S. japonicum* disease is attributed to the ectopic deposition of schistosome eggs, leading to spinal compression or acute encephalopathy. The clinical manifestation of acute schistosomiasis is often observed as acute transversal myelitis or prolonged myeloradiculopathy, with a higher likelihood of affecting the vertebral system [7]. Schistosomiasis infection poses a significant danger to patients having other infectious diseases [55]. HIV-infected persons are also susceptible to schistosomiasis infection. Females with genital schistosomiasis report neovascularization, inflammation, and deformability of the genital epithelium. They may also have a damaged physical barrier that makes them more vulnerable to HIV infection during sexual activity. A high intensity of *S. haematobium* infection increases CD-4 positive cells in the semen of males. Schistosomiasis could also lead to perturbation in the immune system towards infections, allergens, or vaccinations co-infecting each other. It is observed that during infection, T-helper-1-type immune responses are downregulated [56]. T-helper cells play a vital role and are associated with controlling viral or protozoan infections and with immunization. 

Besides this, the problem of morbidity/co-morbidity is the stigma associated in many SSA countries with both HIV and, in some societies, gynecological symptoms similar to sexually transmitted infections, with schistosomiasis [57].

Isaiah et al. (2022) adopted a five-stage scoping review process of identifying research queries; identifying relevant studies; charting data; and collating, summarizing, and reporting results to find the epidemiology of schistosomiasis in children below 6 years residing in hard-to-reach locations. The burden of schistosomiasis among marginalized preschool-aged children was studied to generate evidence on the requirement for the addition of this population when designing the expansion of PCT programs [58].

## 6. Recent Outbreaks of Schistosomiasis

Recently, there have been several outbreaks of schistosomiasis in various countries around the world, varying from mild to severe intensity. Kayuni et al. studied the outbreak of ItS alongside increasing UGS occurrence in primary-level SAC living in Malawi’s Mangochi district shoreline. About 520 children in lakeshore primary schools were investigated for *S. mansoni* by using the positive urine circulating cathodic antigen (CCA) dipstick method. By applying the above techniques, the mean prevalence was 31.5% (95% confidence interval (CI): 27.5–35.5) and 24.0% (95% CI: 20.3–27.7) for UGS for *S. mansoni* [59].

A UGS outbreak caused *S. haematobium* to affect more than 1000 people, out of which the majority were children in the Congo. The alteration in the ecology and an increase in the snail population due to the construction of a hydroelectric dam were the main reasons behind this outbreak [60]. An *S. mansoni*-based ItS outbreak in Ethiopia affected more than 5000 people, out of which the majority were mostly primary-level SAC. The inaccessibility of pure water and sanitation amenities and the limited awareness of the disease among the community were the major factors for this outbreak. An *S. haematobium*-based UGS outbreak in Nigeria affected more than 10,000 people, out of which most were females [61]. The use of contaminated water sources in houses and agricultural fields, along with limited access to health services, were the major factors for this outbreak. An ItS outbreak by *S. japonicum* was observed in China, which affected more than 20,000 people, and the majority of them were farmers and fishermen. It was found that environmental changes and the migration of infected people and animals due to floods and landslides were the major reasons for the outbreak [62].

## 7. Diagnosis of Schistosomiasis

The diagnosis of different *Schistosoma* species is possible by conventional emerging molecular techniques. The proper diagnosis of specific *Schistosoma* spps., along with the type of schistosomiasis, is crucial in providing timely treatment to the patient. Precision, accuracy, specificity, and sensitivity are some of the major issues faced by conventional diagnosis methods. All these hurdles can be overcome by using more reliable and sensitive techniques like molecular methods, CRISPR-cas, nucleic acid tests, antigen tests, serological methods, etc. Most of these methods are highly sensitive and specific and require very small sample amounts, so they can be diagnosed at the acute phase of the disease only and help the patient in providing timely treatment. The gravity of the situation necessitates the use of these advanced and accurate diagnosis methods [63]. All these diagnosis methods are described below in detail. 

### 7.1. Conventional Diagnosis of Schistosomiasis

Conventional diagnostic methods for schistosomiasis primarily focus on detecting the presence of the parasite or its eggs in the human body, utilizing various laboratory methods (microscopy of stool and urine samples, serological tests, rectal or bladder biopsy, hematuria testing for *S. haematobium*, and imaging techniques like ultrasound and computing tomography). All these techniques, with their mechanisms, advantages, and drawbacks, are mentioned below in Table 2. Viable eggs found in urine (*S. haematobium*), stool (*S. japonicum*, *S. mansoni*), or tissue samples are indicators of schistosomiasis. Although urine and feces have limited sensitivity, the existence of infectious schistosomes cannot be excluded.

According to the WHO, urine dipstick tests for heme and microscopic analysis of polycarbonate screens or nylon used for eggs in urine are advised. Using the Kato–Katz fecal test, eggs in patients with ItS can be found in feces specimens. Varied diagnostic strategies like CCA, the miracidium hatching test (MHT), the Kato–Katz technique (KKT), the formol-ether concentration technique (FECT), point-of-care tests (POCTs), and PCR-based techniques are also an option. Moreover, molecular approaches and DNA identification in serum and urine are also considered.

### 7.2. Serological Methods

Serological examinations for the detection of antibodies are practiced for people living in non-endemic or symptomatic travelers. Such techniques are unable to demarcate between current infection and past schistosomal exposure. To map *S. mansoni* in endemic areas, the lateral flow cassette test, which is performed in urine, is thought to be more efficient than the KKT, which has become the most often utilized method due to its quick nature, simple use, and low training requirements. Figure 6 shows the *Schistosoma*-derived immunological identification of human schistosomiasis.

### 7.3. Advancement in the Diagnosis of Schistosomiasis

The identification of parasitological infections, including schistosomiasis, by conventional microscopic methods, is considered economical due to the inexpensive microscope and the requirement of less trained professional handlers. But at the same time, such microscopic-based diagnostic methods are labor- and time-intensive, having inadequate sensitivity for detecting minute infections. Besides this, there is a requirement for centrifugation or filtration processes in order to concentrate the parasite eggs, which may further prolong the diagnosis stage. The standard direct egg detection methods cannot be used to detect the condition early, before the parasite becomes manifest, while oviposition in intestinal as well as urogenital schistosomes occurs at around 4–6 weeks and 90 days following an infestation of cercaria, respectively [71].

Comparative information on conventional and modern diagnosis methods, along with their benefits and drawbacks for the diagnosis of schistosomiasis, has led investigators to provide and recommend innovative, sensitive, and feasible detection techniques for future strategies. The conventional microscopic techniques were found to be an effective method for the detection of schistosomes, and some relevant and effective methods for the diagnosis of schistosomiasis are in demand. These effective techniques are molecular methods like PCR, loop-mediated isothermal amplification (LAMP), and recombinase polymerase amplification (RPA), which can amplify and detect schistosome DNA or RNA from various biological specimens (blood, urine, saliva) or environmental water. The use of point-of-care (POC) tests like rapid diagnostic tests, lateral flow assays, and smartphone-based devices can also be effective for the diagnosis of schistosomiasis. All these methods are rapid and simple and could be utilized for the diagnosis of schistosomiasis at the field level, using antibodies, antigens, or nucleic acids as detection targets. It is also possible to use biomarkers (microRNAs, cytokines, chemokines, etc.) for the diagnosis of schistosomiasis, which may exhibit the host immune response, parasite load, and stage of the disease, which can be measured by an enzyme-linked immunosorbent assay (ELISA), mass spectroscopy, and biosensors [3].

### 7.4. Proteins as Diagnostic Markers for Schistosomiasis

Proteomics and transcriptomics have recently made significant advancements in the field of the diagnosis of parasitic infections like schistosomiasis. Both of these techniques have resulted in the discovery of a variety of schistosomal proteins and other components secreted at several phases of the life cycle of schistosomes, including intriguing diagnostic prospects. An extensive genetic link between *S. japonicum* and *S. mansoni* was revealed from the study findings of transcriptome and proteomic analysis. A study revealed two other *S. japonicum* proteins that were expressed at various phases of its life cycle, SjTs4 and MF3, with the former being a specific tegument protein (TP) and the latter being an egg-shell protein. Both of these proteins could serve as useful alternate diagnostic targets [3].

Recently, micro-RNAs and extracellular vesicle proteins have been used as diagnostic targets for schistosomiasis. *Schistosoma* TPs are found to be a significant candidate for vaccine and diagnostic markers. The discovery of specific TPs as potential diagnostic markers may help to increase the sensitivity of diagnosing schistosomiasis. The adult *S. japonicum* has two TPs (SjPGM and SjRAD23), which were found to be diagnostic indicators for the illness, according to recent reports. Both of these recombinant proteins may be used to assess the efficacy of medication therapies and to differentiate between current and previous illnesses. Proteins may address the main problems with indirect immunological methods. One major drawback with schistosomal miRNA identification techniques is their high cost and the requirement for more field validation [72]. Table 3 summarizes the various techniques, their principles, required samples, advantages, and disadvantages, along with their sensitivity and specificity for diagnosing schistosomiasis. 

### 7.5. Nucleic Acid Test-Based Diagnosis of Schistosomiasis

The scenario of schistosomiasis varies worldwide, particularly in SSA, as a consequence of different active intervention programs. Despite a few disparities, TchuemTchuenté et al. observed that progress appears to be moving in the direction of eventual eradication. It should be emphasized that variations in schistosomiasis prevalence may result from several factors, including the way the disease is diagnosed [79].

Nucleic acid tests have risen to the top of the priority list among the diagnostic techniques for diagnosing parasitic infections, including schistosomiasis. A specific glass or silicon chip is used in the gene chip test as a carrier for many nucleic acid probes. Important data on gene sequences can be obtained from the chip via fluorescence or sample reactions with nucleic acid probes. The gene chip test’s main advantages are its simplicity, high sensitivity, and specificity. Similar to this, certain diagnostic achievements have been reported in earlier investigations utilizing smaller amounts of blood, saliva, and urine; on the other hand, its practicality needs to be assessed. He et al. (2016) highlighted the advancement of nucleic acid detection (section of the target genes) in the diagnosis and prevention of schistosomiasis [80]. Driscolla et al. [81] developed a PCR method based on the SjR2 sequence, which was utilized for the analysis of cercaria distribution and dynamic changes in the mountainous regions of the Sichuan province of China [81]. The advancement in nucleic acid-based diagnostic techniques, PCR, quantitative qPCR, LAMP, and RPA, along with their benefits and drawbacks, was described by a team led by Ullah. Some of the biosensors based on nucleic acid biomarkers were also found to be effective for the diagnosis of helminthic infection [82]. 

### 7.6. PCR and Multiplex PCR

In the traditional method of PCR, schistosomal DNA is obtained from the patient’s stool, urine, or blood samples. The size of the specific nucleic acid segment band of a *Schistosoma* spp. could be measured by using gel electrophoresis once the DNA is extracted. An infection can be confirmed or excluded by determining the presence or absence of a particular-sized DNA band [83].

#### 7.6.1. PCR-ELISA

Schistosomal DNA identification in feces, sera, and plasma has been enabled by the development of PCR-ELISA [84]. This method may offer an assay for the identification of schistosomes at all clinical illness phases. These molecular methods can also be used to diagnose the status of *Schistosoma* in intermediate snail hosts. The multiplex PCR uses multiple (two or more) pairs of primers in each reaction to simultaneously amplify multiple nucleic acid fragments of an infectious agent. In comparison to the equivalent conventional PCR, it shows better efficiency [85].

Wanlop et al. (2022) developed an easy-to-use and effective miracidium hatching technique (MHT) for the preparation of single-genome DNA of *S. japonicum*. Hatching under sunlight was 92.4%; in fluorescent light, 88.0%; and under dark conditions, 4.7%, suggesting sunlight is the most efficient for this protocol. Microsatellite marker genes were successfully amplified from the DNA extracted from a single miracidium, which also established the quality of the single-genome DNA for succeeding applications [86].

#### 7.6.2. Polymerase Chain Reaction in Real Time

A technique called real-time quantitative PCR (qPCR) is utilized to estimate the quantity of PCR products, frequently by including fluorophores in the reaction design. The produced fluorescence signal would be monitored in every amplification cycle following the preparation of the qPCR reaction since electrophoresis is not required to see the bands.

LAMP is a recently developed gene amplification technique that comprises rapidity, simplicity, and high specificity at constant temperature. It has recently been noted as a workable and reasonably priced alternative to traditional PCR for the detection of schistosomal DNA in feces, urine, and sera. As it targets a specific gene for amplification at steady state temp. (60–65 °C) and for 15 to 60 min, LAMP is often a very sensitive and specific test. This is because it utilizes specific primers from both the inner and outside portions. LAMP foregoes electrophoresis, temperature cycling, and heat denaturation of the template.

#### 7.6.3. PCR Recombinase Amplification

Due to its low resource needs, RPA is an isothermal DNA amplification technique that could be used in the field. According to recent studies, the RPA test has been evaluated for its ability to find small amounts of *S. haematobium* and *S. japonicum.* Additionally, the test is quick and only needs low temperatures. Materials for the assay may also be stored at room temperature. Positive reactions are interpreted by applying lateral flow strips, which suggests that the assay has a good chance of being used in the field.

### 7.7. Antigen Test-Based Schistosomal Diagnosis

Recently, de Sousa et al. (2019) suggested the utilization of a special type of antigen test (POC circulating cathodic) for the detection of ItS in a low-endemicity area. Here, the investigators compared the suggested technique with the most commonly used KKT. It was found that POC-CCA exhibits higher positivity rates in comparison to the KKT. The sensitivity of POC-CCA varied when there was an upsurge in the quantity of analyzed Kato–Katz slides and stool specimens, and the range of the variations was 55.6% to 65.7% when comparing 16 Kato–Katz slides to POC-CCA. In comparison to both techniques, the specificity of POC-CCA tends to be higher than sensitivity, ranging from 76.9% to 80.4%. The Kappa coefficient was used to measure the agreement between the two tests, which is generally weak in low-endemicity areas. From the study, it was concluded that even though the POC-CCA is a rapid and simple diagnostic tool, it has limitations in its performance. A set of Kato–Katz examinations is needed to improve sensitivity and achieve reliable diagnosis in low-endemicity areas. In addition to this, POC-CCA should be used with caution in such settings, especially for programs aiming to eliminate schistosomiasis [87].

Hoekstra et al. (2022) reported that the detection of eggs of parasites in the stool and urine samples has very little sensitivity, so alternative methods could prove to be valuable and do not even need a microscope for examination. The investigation was carried out on 314 banked fecal and urine specimens by using RT-PCR, the POC-CCA test, and the up-converting particle lateral flow circulating anodic antigen (UCP-LF CAA) assay. In urine samples, schistosomal DNA was 3%, while in fecal specimens, it was 28%. By applying CCA and circulating anodic antigen (CAA), DNA was about 28% and 69%, respectively. Further analysis exhibited that fecal-based PCR and the POC-CCA test are the most appropriate diagnostic methods for screening *S. mansoni* disease without the need for a microscope. Table 4 shows *Schistosoma*-specific nucleic acid targets and their importance in the molecular diagnosis of human schistosomiasis.

### 7.8. CRISPR/Cas13a-Based Assay for Detection of Schistosomiasis

It is one of the most recent techniques in the field of diagnosis of diseases related to schistosomes. Several attempts were made for the CRISPR/Cas 13-based diagnosis of schistosomiasis. Recently, MacGregor et al. developed a CRISPR-based detection platform called SHERLOCK (specific high-sensitivity enzymatic reporter unlocking) for the identification of *S. japonicum* and *S. mansoni* by combining RPA with CRISPR-Cas13a detection. The detection and measurement of the sample were based on fluorescent and colorimetric methods. The evaluation of the SHERLOCK assays was performed by using fecal and serum samples from infected mice and humans. The fecal/serum samples were collected from 150 ARC Swiss female mice (infected with *S. mansoni*) and 189 humans (infected with *S. japonicum*—endemic regions in the Philippines—and *S. mansoni*—endemic regions in Uganda). Investigators found that the *S. japonicum* and *S. mansoni* SHERLOCK assays showed high sensitivity and specificity in detecting *Schistosoma* infections in both animal and human samples (93–100%) in comparison to the gold-standard qPCR. Finally, it was interpreted that SHERLOCK/CRISPR-based diagnostics of schistosomiasis demonstrate very great potential and are highly accurate. Moreover, such techniques are also field-friendly POC assays that may help in providing the next generation of diagnostic and surveillance apparatuses for such types of illnesses [121].

Cherkaoui et al. (2023) fabricated a CATSH (CRISPR-assisted diagnostic test for *S. haematobium*), which uses RPA, Cas12a-targeted cleavage, and portable real-time fluorescence identification. Such CRISPR-based diagnosis was highly sensitive and specific for the identification of infectious parasites in remote locations, which could play a vital role in the control and eradication of such illnesses. The fabricated device has a high analytical sensitivity, consistency in the identification of single eggs of the parasite, and specificity for UGS-causing species. The device uses simulated urine samples (having parasitic eggs) and provides results within 2 h.

### 7.9. Point of Care (POC) and Other Emerging Methods

POC diagnostics of schistosomiasis could be mainly categorized into three classes, namely, immunological POCTs, molecular-based POCTs, and mobile phone microscopes. Immunological POCTs could be further categorized into antigen detection (AgD) and antibody detection (AbD). AgD is based on the detection of *Schistosomal* spp. proteins and DNA in biological specimens (blood and urine). The principle of such tests relies on the identification of the presence of parasites by directly detecting particular antigens produced by *Schistosoma*. The AbD test identifies the presence of antibodies developed against *Schistosoma* in biological specimens (blood or urine). Such a test suggests the previous exposure of an individual to the parasite and the subsequent development of an immune response. AgD-based methods further include POC-CCA, Eco Teste (POC-ECO), and UCP-LF-CAA. AbD-based methods include the dipstick dye immunoassay (DDIA) [122], gold immunochromatography assay (GICA), and immunochromatography test (ICT) [3], which are shown in Figure 7. Molecular-based POCT diagnostics include LAMP and RPA in addition to lateral flow immunoassays, where the best technique based on LAMP is SHERLOCK, while CSTSH is the best molecular method for RPA. Molecular-based POCT diagnostics are a highly reliable and widely developed approach for the screening of schistosomiasis. The final method, i.e., the mobile phone-based method, includes SchistoScope (Meulah et al., 2022), CellScope (Armstrong et al., 2022), and FoldScope, [123], which have shown several advancements in the diagnosis and screening of schistosomiasis. Due to the latest advancements, it has been possible to develop molecular-based POC diagnostics for schistosomiasis. Using artificial intelligence with smartphone-based optical devices, such as the SchistoScope, could provide an economical and easy-to-deploy POC diagnostic for UGS. Figure 7 summarizes the current and upcoming POC diagnostic tests for schistosomiasis. Table 5 shows LFIAs for the diagnosis/screening of schistosomiasis.

It is concluded that the new methods are promising and important because women with FGS may not excrete schistosome eggs. In addition to this, health education also plays a major role in the diagnosis of schistosomiasis by promoting awareness, prevention, and early detection. It helps communities understand the disease, recognize symptoms, and seek timely medical care, reducing severe complications. Education also encourages preventive measures like avoiding contaminated water and practicing good hygiene. Public campaigns highlight available diagnostic procedures, encouraging early testing and treatment. Health education supports community-based screening programs and informs individuals about diagnostic tools, increasing early detection. Finally, it helps post-diagnosis management, ensuring adherence to treatment and reducing the spread of infection.

## 8. Prevention and Control of Schistosomiasis

The prevention strategy includes avoiding coming in an encounter with fresh water that is parasite-infected, having access to clean water, maintaining decent hygiene, and controlling snails. Drugs do not cure FGS and need to be prevented. As per the WHO strategy, groups targeted for treatment are categorized into pre-SAC and SAC, as well as adults associated with fishing, farming, and irrigation who have frequent exposure to infested water residing in highly endemic areas [134]. The WHO strategies include PZQ and also large-scale treatment with PCT. The treatment of preschool children is recommended by the WHO, as well as the routine treatment of populations at risk, which will ease minor clinical signs and stop infected individuals from acquiring severe, late-phase chronic illnesses. As per the latest WHO reports for 2021, globally, there was a requirement for treatment for about 29.9% of the patients. About 43.3% of school-aged children required PCT for schistosomiasis. For the control of schistosomiasis, the WHO strategy focuses on minimizing illness through periodic, targeted treatment with PZQ through the large-scale PCT of affected individuals. According to WHO 2021 reports, due to COVID-19, there was a decrease of 38% in comparison to 2019 due to the suspension of treatment campaigns in several endemic areas. Recently, in 2022, Kokaliaris and their team carried out a spatiotemporal modeling study on the PZQ-based effect of PCT on schistosomiasis among SAC in SAA [135].

Exosomes not only have a significant impact on the pathophysiology of schistosomiasis, but they may also play a role in the regulation of the host immune response, which could prove to be a new direction for controlling and preventing the disease. Exosomes are important mediators for intercellular communication (facilitating the exchange of proteins, lipids, and DNA/RNA between cells), and they are associated with the progression of schistosomiasis and could play a critical role during host–parasite interactions [136]. 

## 9. Treatment of Schistosomiasis

PZQ is the most recommended medicine against schistosomiasis, which is less active against juvenile parasites than mature parasites [137]. In order to exhibit better efficacy, it requires a better antibody response, and if there is poor efficacy, then it is a major issue for premature schistosome larvae. PZQ has been the primary treatment for schistosomiasis for over four decades, showing high efficacy with a single dose of 40 mg/kg body weight for most Schistosoma species in all age groups, as recommended by the WHO. However, a dose of about 60 mg/kg is suggested mainly for *S. japonicum* and *S. mekongi* [138]. Cure rates (CRs) range from 57% to 88%, and egg reduction rates are consistently high at 95% across various schistosome species and patient demographics [139]. The usage of PZQ may exhibit some common side effects like gastrointestinal discomfort, headache, light-headedness, and infrequent blood in the stool. The risk of side effects may be seen only in patients having a higher intensity of infection. Besides this, pharmacogenetic variations, like CYP2C19 genotypes, can influence PZQ plasma concentrations and treatment outcomes, suggesting that a single dose may not be universally effective [140].

Besides PZQ, antimalarial artemisinin derivatives are also effective against immature larval forms of schistosomes. One major drawback of this drug is that it is not known how much time a cercaria has to be exposed to the drug for enhanced effectiveness. However, a combination of PZQ+ antimalarial artemisinin has shown higher effectiveness and a rate of cure against schistosomiasis [141,142]. One study used a combination of PZQ with dihydroartemisinin-piperaquine (DHP), which demonstrated superior effectiveness, significantly increasing systemic PZQ exposure and potentially improving treatment efficacy [143].

Furthermore, snail management could also be one of the preventive measures to lower the transmission of schistosomiasis, which is possible by using linalool or *Cinnamomum camphora* (L) extracts. Linalool-treated snails developed gill disintegration and cell degradation, so it is also used for the treatment of *S. japonicum* infection [144]. Pham et al. [145] studied about 100 participants with *S. mansoni* infection in Tanzania for six months after treatment with PZQ by using a cohort study. The effect of PZQ on the infection and lower gastrointestinal mucosa (LGIM) was evaluated by using sigmoidoscopy, histopathology, and stool tests. It was found that *S. mansoni* infection was associated with mild-to-moderate LGIM abnormalities that were grossly visible during sigmoidoscopy. There was a partial improvement in these changes nearly six months after effective treatment with PZQ [145].

Recently, several attempts were made in which instead of a single drug, a combination of drugs was administered to the patient against schistosomiasis. For instance, in a study, PZQ + moxidectin was given against both *S. mansoni* and *S. haematobium* infections [146]. Both drugs have different modes of action against parasitic worms, where the former damages the worm’s tegument, which exposes it to the host’s immune system and leads to its expulsion, while the latter interferes with the worm’s nervous system, causing paralysis and death. The combined regime was well tolerated and effective, and it was found that such a combination could offer a broader spectrum of activity against different *Schistosoma* species. In this study, 173 investigated participants (from Côte d’Ivoire) who were having infections of both *S. mansoni* and *S. haematobium* were given these two drugs. Each patient was given a single dose of PZQ and moxidectin in the amount of 40 mg/kg and 8 mg, respectively. The primary outcome was the CR of *S. mansoni* infection 12 weeks after treatment. The secondary outcomes included the CR of *S. haematobium* infection, the reduction in egg counts, and the safety and tolerability of the combination. The result showed that the combined dose was much more effective than PZQ alone in curing both *S. mansoni* and *S. haematobium* infections. The CR of *S. mansoni* infection was 92.9% in the combination group, compared to 76.2% in the PZQ group. The CR of *S. haematobium* infection was 100% in the combination group, compared to 89.7% in the PZQ group. The combination also reduced the egg counts of both species by more than 99%. It was concluded that the combination of these two drugs could be a valuable tool for the control and elimination of schistosomiasis, especially in areas where both *S. mansoni* and *S. haematobium* coexist [135]. 

In another study, artesunate-based regimens were used for the treatment of schistosomiasis, where a promising result was obtained for artesunate–PZQ against *S. mansoni* [147]. In another study, artesunate–mefloquine was effectively used for the treatment of *S. haematobium* infection [148]. 

Several investigators suggested alternative delivery methods of the current drugs for their enhanced effect. For instance, in a study, PZQ was given intradermally to a patient. The result showed good tolerability and potential for improved adherence compared to oral administration. In another study, investigators used long-acting injectable (LAI) PZQ formulations and suggested that such formulations could simplify treatment regimens and improve adherence by providing sustained drug release over extended periods, particularly in mass drug administration programs [149]. LAI formulations are designed to release the active pharmaceutical ingredient over a prolonged period, reducing the frequency of administration. This is particularly beneficial for chronic conditions requiring long-term treatment, as it can enhance patient compliance and reduce the risk of relapse [150,151]. By decreasing the dosing frequency, LAIs simplify treatment regimens, which can lead to better adherence. This is crucial in managing diseases like schistosomiasis, where consistent medication intake is necessary to prevent infection and complications [150]. One of the studies carried out by Lei et al. exhibited that in PZQ implants for schistosomiasis, stable plasma concentrations were maintained for up to 70 days, effectively preventing *S. japonicum* infection in mice during early infection periods. The implants maintained effective drug levels in the bloodstream, preventing infection when administered before or shortly after exposure to the parasite [151].

Few investigators optimized the existing regimens of the drugs against schistosomiasis, like single versus multiple PZQ doses, where the investigators compared the efficacy and economics of single versus multiple PZQ doses for different *Schistosoma* spp. and infection intensities. Increasing the dose to 80 mg/kg or administering multiple doses can enhance treatment outcomes, particularly in preschool-aged children, where the standard dose may be insufficient [152]. Some of the investigators optimized the treatment timing and integration with other interventions (water sanitation and hygiene programs). Research is exploring the optimal timing of treatment for schistosomiasis, as well as its integration with other interventions, to maximize impact and prevent reinfection [153]. In some of the most recent attempts, new treatment options were explored against schistosomiasis, which includes synthesizing next-generation PZQ derivatives for the treatment of schistosomiasis [154]. Such modified versions of the PZQ have improved efficacy and potentially longer-lasting effects against various types of schistosomes. Table 6 provides summarized information on all the treatments evaluated, used in combination with their role in the treatment of schistosomiasis. 

Health education is essential for diagnosing and treating schistosomiasis, especially with PZQ and novel therapies. Communities can better comply with PZQ therapy by boosting awareness of its efficacy, doses, and negative effects. Education promotes early diagnosis through stool and urine tests and timely medical assistance. Understanding combination treatments like PZQ and antimalarial drugs can improve acceptance and efficacy. Community education on snail management and cleanliness strengthens transmission prevention. Awareness of vaccination research boosts clinical trial trust and participation. Finally, integrating schistosomiasis treatment with broader health initiatives, like water sanitation programs, can significantly reduce reinfection rates and improve overall health outcomes.

## 10. Vaccine Development for Schistosomiasis with Possible Targets and Clinical Trials

To date, several vaccines have been manufactured for the control and prevention of various forms of schistosomiasis around the world. For the development of the vaccine, two major strategies are involved: firstly, the identification of target sites on the Schistosoma for the development of the vaccine, and second, the development of a method for the delivery of the vaccine. To date, several methods have been employed for the identification of target sites on the surface of *Schistosoma*, for instance, gene editing [156], CRISPR/Cas, proteomics, immunomics, exosomics, gene silencing, etc. The techniques that could be applied for the delivery of a *Schistosoma* vaccine are a DNA vaccine, an irradiated cercaria vaccine, an epitope, and a recombinant vaccine. Some of the vaccines which are under development and clinical trials are discussed below in brief. 

Molehin et al. (2022) emphasized that the continuous use of a single anthelmintic PZQ drug may lead to drug resistance in the parasites, so some new possibilities must be explored, like vaccines. So, investigators provided the recent progress in the field of schistosome-based vaccine development, including the Sm-p80 vaccine, which has shown efficacy in preclinical trials. The schistosomal vaccine utilizes cutting-edge techniques like the development of mRNA vaccines and CRISPR-based technology exploitation to deliver valuable and innovative understanding into future vaccine design, discovery, manufacturing, and deployment. In a significant study, anti-schistosomal vaccines were developed against parasites, showing promising potential for the long-term prevention and control of the illness. 

Different *Schistosoma* species present a variety of structures on their surface that could serve as potential candidates for vaccine development. Additionally, there are several proteins and other antigenic structures, such as enzymes, that may play a crucial role in schistosomal vaccine development. Notable among these are the 28 kDa glutathione S-transferase (*S. haematobium*) [Sh28GST], the tetraspanins TSP-1 (Sm-TSP-1) and TSP-2 (Sm-TSP-2) (*S. mansoni*), calpain (Sm-p80) (*S. mansoni*), and a 14 kDa fatty acid-binding protein (FABP) (Sm14) (*S. mansoni*) [Sm14 FABP].

### 10.1. S. haematobium 28 kDa Glutathione S-Transferases (Sh28GST)

Sh28GST is an enzyme that is expressed on the surface of the schistosome, which is engaged in the uptake, transport, and compartmentalization of host-derived sterols and fatty acids in addition to parasite detoxification pathways. It is known for abrogating the movement of epidermal Langerhans cells to the draining lymph nodes, in addition to the functional specific binding of testosterone. It is also involved in prostaglandin D2 synthesis, which contributes a major role in parasite immune evasion. To date, this enzyme has been extensively assessed as a potential vaccine against schistosomiasis in different animal models. Several investigations were carried out in different primates and animals where recombinant proteins were expressed in the *Saccharomyces cerevisiae* and exhibited a significant level of reduction in the *Schistosoma* burden, i.e., 40–60%. Moreover, a significant decrease in female worm fecundity and egg viability was noticed. Patas monkeys were injected with the rSh28GST protein, where the recovery rate was just 4% from the adult worm infection, indicating that the Patas monkeys were not the preferred host for the infection of *S. haematobium*. The rSh28GST protein injection could have a significant impact on tissue egg load and stool egg excretion, which suggests its potential effectiveness in reducing the spread of the parasite [157].

During a phase I trial, safety and tolerability were investigated in juvenile, healthy, Caucasian male adult vaccinees after injecting 2–3 intravenous injections of 100 µg rSh28GST antigen + alum (adjuvant). Clinical trials assessed the safety, tolerability, and immunogenicity of the above-developed vaccine among adults and children residing in endemic areas [158]. The developed vaccine was safer and highly immunogenic in adults, where it induces interleukin (IL)-5 and IL-13. Besides this, IgE was completely absent, and IgG1 antibodies were predominant, which could potentially inhibit the enzymatic property of the immunogen [159]. Out of all the vaccines of schistosomes, the based antigen has reached the third phase of clinical trials. During the third phase, this vaccine was given subcutaneously to 250 Senegalese children in the 6–9-year-old age group. The vaccine-administered recipient children were categorized into two classes, i.e., (i) the Bilhvax group and (ii) the controlled group. The former group received three subcutaneous injectable doses of rSh28GST/Alhydrogel after 0, 4, and 8 weeks and a booster dose at 52 weeks (almost 1 year after the first injection), while the latter group received only alhydrogel followed by PZQ treatment after 44 weeks based on previous phase II results. Further, the efficacy of the vaccine was assessed based on a delay in the recurrence of UGS, where the recurrence was marked by the presence of microhematuria (blood in urine) along with at least one living parasite egg in urine from the baseline to week 152. During the follow-up time of 152 weeks, no remarkable variations in the incidence of serious adverse events between the study arms were observed. Both groups exhibited similar rates of recurrence after 152 weeks: 108 children in the Bilhvax group and 112 in the control group. During the examination of the protective nature of the vaccine, it was found that the recipient of the vaccine (rSh28GST) exhibited higher levels of specific IgG1, IgG2, and IgG4 antibodies but a lack of IgG3 and IgA isotypes. In humans, acquired immunity against schistosomiasis was due to IgG3 and IgA antibodies, whereas this vaccine was found to be ineffective against UGS [159]. The vaccine is currently undergoing further testing to determine its efficacy in other regions. 

### 10.2. S. mansoni Tetraspanin (Sm-TSP-2)

Tetraspanins (TSPs) are a family of proteins that are mainly presented on the surface membrane of schistosomes, and hence, these are exposed to the host immune system. Structurally, tetraspanins are formed of four transmembrane domains linked by two extracellular loops, out of which one is shorter and one is longer. *S. manoni* has mainly two types of tetraspanins: TSP-1 (Sm-TSP-1) and TSP-2 (Sm-TSP-2). The latter protects the 40% benchmark set by the WHO for the development of schistosome vaccine antigens in clinical trials [160]. In one study, mice were injected with the rSm-TSP-2 protein, which led to a reduction in the burden of worms by up to 57% and a burden of liver eggs by up to 64% in comparison to untreated mice [161].

The Sm-TSP-2/Alhydrogel vaccine was given to adults under a randomized, controlled phase Ib trial in a region having ongoing *S. mansoni* outbreaks. The vaccine was safer, minimally reactogenic, and induced significant IgG and IgG subclass responses against the antigen. Based on these promising results, a phase II trial is underway in an endemic area of Uganda [162].

### 10.3. Schistosoma mansoni Calpain (Sm-p80)

Calpain (Sm-p80) is a calcium-activated neutral protease that is presented in abundance on the tegument of adults and several life cycle stages of parasites. It is a promising vaccine candidate for schistosomiasis. Sm-p80, a subunit of *S. mansoni* calpain, is crucial for the survival of parasites in their host environment, making it a potential vaccine candidate for schistosomiasis. An investigation performed on mice exhibited that the Sm-p80 vaccine provided 70% protection against schistosomiasis [163]. Currently, this vaccine is ready for human clinical trials. In a double-blind experiment, baboons were vaccinated with Sm-p80/GLA-SE (adjuvant), which reduced 93.45% of adult female worms with a consequent decrease in tissue egg load by nearly 89.95% [164]. Additionally, the hatching rate of eggs excreted in the stool also decreased significantly. In another study, baboons were firstly treated with PZQ, followed by vaccination with Sm-p80, which resulted in a significant decrease in the quantity of parasite eggs retained in the tissues of the baboons. Moreover, in the vaccinated baboons, there were lowered hatching rates of parasites in comparison to the non-vaccinated ones.

### 10.4. S. mansoni 14 kDa Fatty Acid-Binding Protein [FABP] (Sm14)

Sm14/FABP is a protein that is present on the surface of the schistosome and is involved in the uptake, transport, and compartmentalization of fatty acids and sterols derived from the host. They do not have oxygen-dependent pathways, which is one of the promising candidates for the manufacturing of a vaccine for anti-schistosomiasis. In one investigation, Sm14 protein-based vaccines were given to outbred Swiss mice; as a consequence of this, there was a remarkable decrease in the adult worm burden in the mice, i.e., 67%. This investigation evidenced that Sm14 has a strong protection effect against the parasites. Further, upon the administration of the Sm14 protein in New Zealand white rabbits after infection with *S. mansoni* cercariae, it exhibited up to 93% protection. So, this suggests strong protection against schistosomiasis in animal models [165]. 

In a recent investigation, authors have reported successful phase I trials of the newly developed vaccine “Sm14 + glucopyranosyl lipid A in squalene emulsion (GLA-SE)” in a ratio of 50 µg/10 µg, which was administered intramuscularly at a rate of three per dose over 30-day intervals. In phase Ia trials, pregnant rabbits were vaccinated, and results showed no changes in their pregnancy parameters. No toxicological features was observed in the mother rabbits and the offspring, which means the vaccine can stimulate a protective immune response. Further, another phase Ib trial was also conducted on healthy young adult women, where no adverse effect of the vaccine was noticed. The vaccine causes an elevation in IgG antibody levels that contributes a crucial role in immune defense against infections. Besides this, all the vaccines showed an elicited robust cytokine response, with increased TNF-α, IFN-γ, and IL-2 profiles on the 90th and 120th day. This ensures the vaccine does not cause any harmful side effects or complications in humans and thus was suggested for a phase II trial [166]. The vaccine developed from this antigen was used for a phase I clinical trial in Brazil. The vaccine was safer and well tolerated, which also induced a significant increase in anti-Sm14 antibody levels in the vaccinated candidate. Currently, it is under a phase II clinical trial in Brazil. Detailed information about the involvement of techniques for the development of vaccines against schistosomiasis is described below in Table 7.

Besides this, proteomics, exosomics, and immunomics also may play important roles in the formulation of vaccines against schistosomiasis. For instance, exosomes, a subset of extracellular vesicles, are involved in cell-to-cell communication and have been explored for their potential in vaccine development due to their ability to deliver a wide range of molecules, including proteins, lipids, and nucleic acids, to target cells. This capability makes them suitable candidates for novel vaccine platforms, particularly against complex pathogens like *Schistosoma* spp. [167]. They are preferred in vaccines due to their high biocompatibility and reduced immunogenicity. Besides this, they have the potential to cross the blood–brain barrier, which further enhances their candidature as a vaccine delivery system. From the studies, it was found that exosomes can deliver their cargo over long distances, mediating various biological functions. This property is particularly useful in vaccine development, where the targeted delivery of antigens is crucial for inducing a robust immune response [168]. Recent advancements in exosome engineering, such as surface modification and cargo loading, have enhanced their targeting abilities and therapeutic efficacy. These innovations are crucial for developing effective vaccines against complex pathogens like *Schistosoma* spp. [169,170].


tropicalmed-09-00243-t007_Table 7Table 7Various technologies applied for the manufacturing of anti-schistosomal vaccines.Techniques Used for the Identification of Vaccine TargetsExamples of Methods AppliedReferencesGene editingCRISPR/Cas-9[171]Transcriptomics and DNA microarray profiling RNA sequencing, next-generation sequencing [172]Proteomics Reverse vaccinology approach, proteasomal cleavage and TAP transport prediction, epitope prediction, 3D structure prediction and refinement[173]Exosomics-[136,168]ImmunomicsELISPOT, immunomic microarray, mapping tools for the epitopes of T and B cells[174]Immunoinformatics Multi-epitope peptide-based, transmembrane proteins as a target[173]Gene suppression iRNA, vector-based silencing, lentiviral transduction[175]
**Techniques used in vaccine delivery**
Antibody and chromatography-based techniques 
DNA-based vaccines SjCPTI, Smp80[175]Irradiated cercarial vaccineCulturing of cercariae, followed by irradiation[176]Synthetic multiple epitope peptides Sm14[177]Epitope based vaccine Transmembrane proteins, codon optimization for *E. coli* to ensure heterologous expression and antigen purification, alongside stability and solubility prediction[173]Recombinant protein vaccines or bivalent vaccines Smp80, Sm97, Sm14 (paramyosin), Sm-TSP-2, Sm14/Sm29, Sm14/Sm-TSP2/Sm29/Smp80[178]New adjuvants R848, TLR7/8 agonist, CpG-ODN, QuilA, GLA-SE, alum, poly (I: C)[175]


Health education is essential for the advancement and implementation of schistosomiasis vaccines by enhancing community awareness and engagement in vaccination initiatives. Understanding potential vaccine targets, including specific proteins from various *Schistosoma* species—such as the 28 kDa glutathione S-transferase (Sh28GST), tetraspanins (Sm-TSP-1 and Sm-TSP-2), calpain (Sm-p80), and the 14 kDa fatty acid-binding protein (Sm14)—can improve the acceptance of these vaccines. Health education can explain how these antigens work and their benefits. This may encourage participation in clinical trials and boost public confidence in vaccine safety and effectiveness. Awareness of current vaccine development can encourage individuals to adopt preventative steps and demand vaccines, which may result in improved health outcomes. Moreover, educating populations on the importance of early diagnosis and treatment adherence enhances vaccination efforts, forming an integrated strategy to effectively address schistosomiasis.

## 11. Vaccine Candidate in Experimental Trials

### 11.1. Surface Membrane Candidate Vaccines (Sm23)

Surface membrane 23 kDa (Sm23) is an integral membrane protein exposed on the apical membrane of the parasite. Numerous efforts have been made for the development of an effective vaccine using Sm23 in plasmid DNA (pcDNA), multiple antigenic peptides, and recombinant ® vaccine constructs. It has been found that a limited number of conformational epitopes on Sm23 and other tegmental proteins can elicit mouse, rat, and human production of serum antibodies against *S. mansoni* infection. It was found that it failed to influence schistosomula or adult worm survival, suggesting that there is a need to re-evaluate host immune responses to many schistosome antigens. This parasite has ESP or, in EVs, exosome-like, 120k pellet vesicles and microvesicle-like, 15k, which may act as a reservoir for different vaccine candidates like TSPs Sm23, Sm-TSP-1, and Sm-TSP-2. This is how it shows protection [179]. 

### 11.2. Glucose Transporter Proteins (GTPs)

The adult schistosome acquires its glucose from the blood of the host. The *S. mansoni* has two different glucose transporter proteins (GTPs)**,** SGTP 1 and SGTP 4 antigens, which are present in the tegument. The former is expressed in the tegmental basal membrane and other tissues of different life stages of the schistosome, while the latter is present in the host interactive apical tegmental membranes [56]. Moreover, the latter facilitates the import of glucose from the host bloodstream into the tegument. Parasites lacking these two GTPs are unable to import glucose, which provides evidence for the importance of these SGTPs in importing exogenous glucose and then affecting parasite development in the mammalian host. In all experimental trials, the GTP-based vaccine formulation showed a small amount to no protection against *S. mansoni* in laboratory animals, which might be due to no physical membrane antigen being accessible to host antibodies [180]. The current status of all the schistosomal vaccines that are under clinical trials is presented in Table 8.

From the above section, it was found that there are more vaccine candidates for *S. mansoni* than for *S. haematobium*, which can be attributed to several factors, including differences in research focus, biological characteristics, and technological advancements. *S. mansoni* has been the primary focus of vaccine research due to its widespread prevalence and the availability of advanced research methodologies that have facilitated the identification of numerous vaccine candidates. This focus has resulted in a more extensive pipeline of potential vaccines for *S. mansoni* than for *S. haematobium*. *S. mansoni* is the most widespread schistosome species infecting humans, accounting for a significant portion of global cases, especially in Africa, where 90% of infections occur. Its high prevalence has led to greater research focus and the development of multiple vaccine candidates, with several, such as Sm14, SmTSP-2, and Sm-p80, advancing to clinical trials [181,182]. Advances in technologies like genomics, transcriptomics, and proteomics have greatly enhanced the identification of *S. mansoni* vaccine candidates by uncovering antigens and epitopes that elicit protective immune responses. High-throughput screening methods, such as phage display and immunoinformatics, have further aided in identifying numerous potential targets, including those in the parasite’s digestive tract and tegument proteins [183,184,185]. Extensive studies on *S. mansoni*’s biological complexity and host–parasite interactions have identified proteins in the digestive tract and tegument as promising vaccine targets due to their roles in nutrient uptake and immune evasion. Animal models like mice and primates have facilitated the testing of these candidates, allowing researchers to assess their immunogenicity and efficacy [186,187]. Some *S. mansoni* vaccine candidates, like Sm-p80, have shown cross-species protection against *S. haematobium*, suggesting that research on *S. mansoni* could indirectly benefit the development of vaccines for *S. haematobium* [188]. 

Despite the progress in *S. mansoni* vaccine development, challenges remain, such as the inability to achieve sterile immunity and the need for vaccines that can reduce morbidity rather than completely prevent infection [187,189]. The complexity of the schistosome life cycle and the variability in immune responses among different hosts add to the difficulty of developing a universally effective vaccine [189].

Besides this, from the above section, it is also concluded that it is very important that communities understand how vaccines work and their benefits to ensure uptake when eventually they become available. The backlash against the COVID-19 vaccine emphasizes the importance of promoting new technologies and developing the population’s confidence in their use.

## 12. Challenges in Treatment and Vaccine Development for Schistosomiasis

During the pandemic era, various tropical diseases, including schistosomiasis, were neglected, which led to an increase in the PCT for schistosomiasis from 45.7% to 51.2% in 2019 and 2020, respectively. Moreover, COVID-19 intervened in the delivery of chemotherapy treatment to most remote areas, especially in the African subcontinent. The accessibility and availability of PZQ as the sole drug for schistosomiasis therapy remain a major challenge, mainly among school-going children and pregnant and lactating women, who are not provided with a mass number of drugs. The elimination and control of schistosomiasis from other sectors like agriculture, education, and the environment need more effective measures to reduce the transmission of the disease. The non-availability of a single approved vaccine for schistosomiasis is another major concern, as the continuous use of a single medicine may result in drug resistance among patients.

## 13. Conclusions

Even in the 21st century, schistosomiasis is one of the major parasitic infections that is widely spread in the tropical regions of various continents due to negligence. A large population is affected due to the non-availability of medicine or vaccines, clean water, sanitation, etc. The severity of the disease can vary from person to person, but school children, farmers, and pregnant and lactating women are more prone. Moreover, environmental factors like water contamination, disturbances in water ecology, etc., have a significant role in disease occurrence and outbreak. Molecular methods have been found to be more accurate for the diagnosis of schistosomiasis in comparison to the conventional method. The combination of drugs and a regime is more effective for schistosomiasis therapy in comparison to praziquantel alone. Thorough knowledge of schistosomiasis diseases among endemic areas is recommended to achieve the complete eradication of schistosomiasis. 

## Figures and Tables

**Figure 1 tropicalmed-09-00243-f001:**
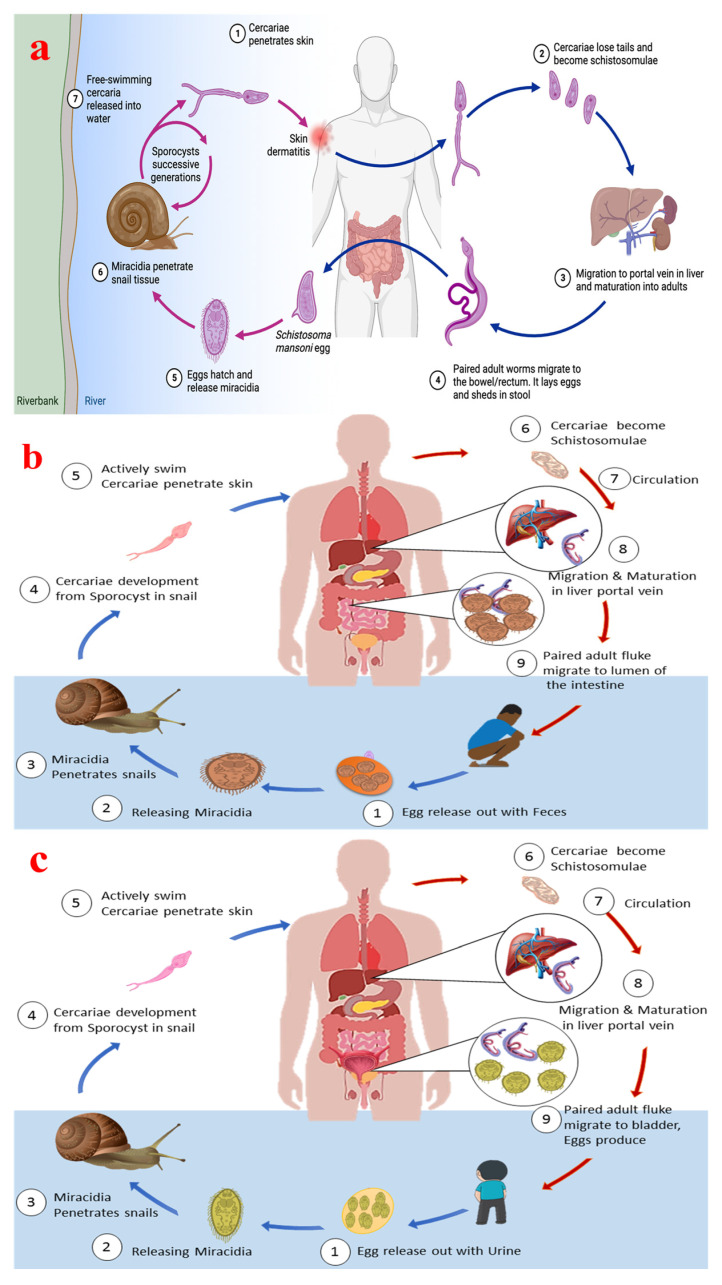
(**a**) Schematic infection cycle of *schistosomiasis* in human hosts, (**b**) the life cycle of *S. japonicum* and intestinal schistosomiasis, and (**c**) the life cycle of *S. haematobium* and resulting genital schistosomiasis. (**a**) was produced with BioRender (www.biorender.com; accessed on 8 January 2024).

**Figure 3 tropicalmed-09-00243-f003:**
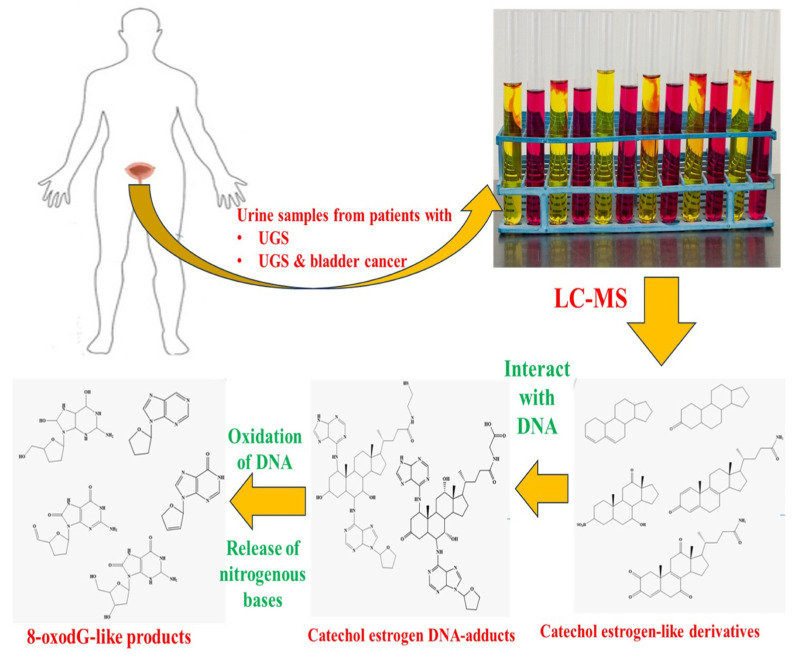
The metabolites present in urine with urogenital schistosomiasis and UGS-associated SCC and absent from the urine samples of healthy individuals. The figure is adapted from Santos et al. (2021) [37], Bruner et al. (2000) [39], and Gouveia et al. (2020) with permission, along with license no: 5887131166730 [40].

**Figure 4 tropicalmed-09-00243-f004:**
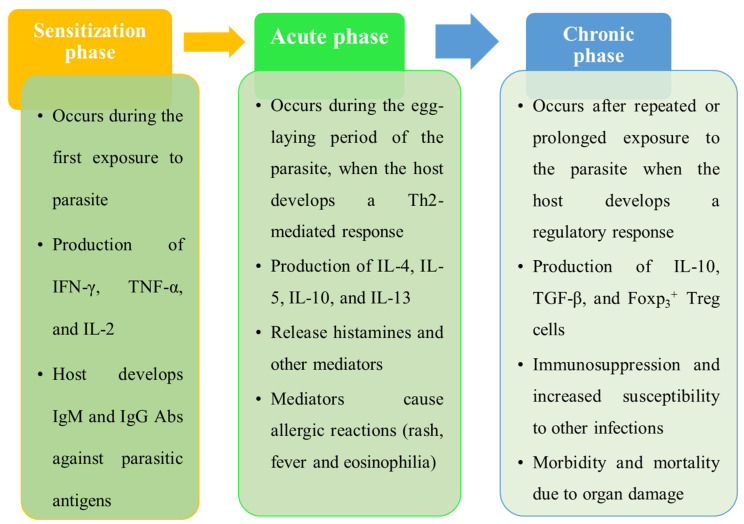
Different phases of immune response in schistosomiasis.

**Figure 6 tropicalmed-09-00243-f006:**
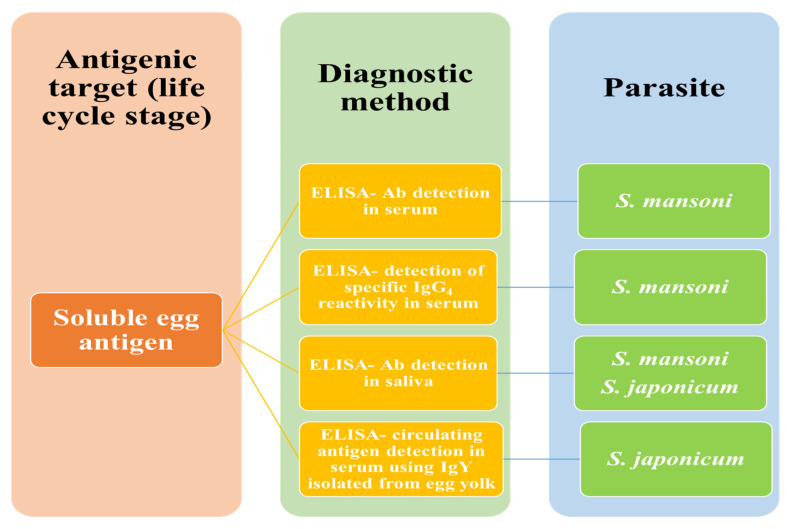
*Schistosoma*-derived immunological diagnosis of human schistosomiasis.

**Figure 7 tropicalmed-09-00243-f007:**
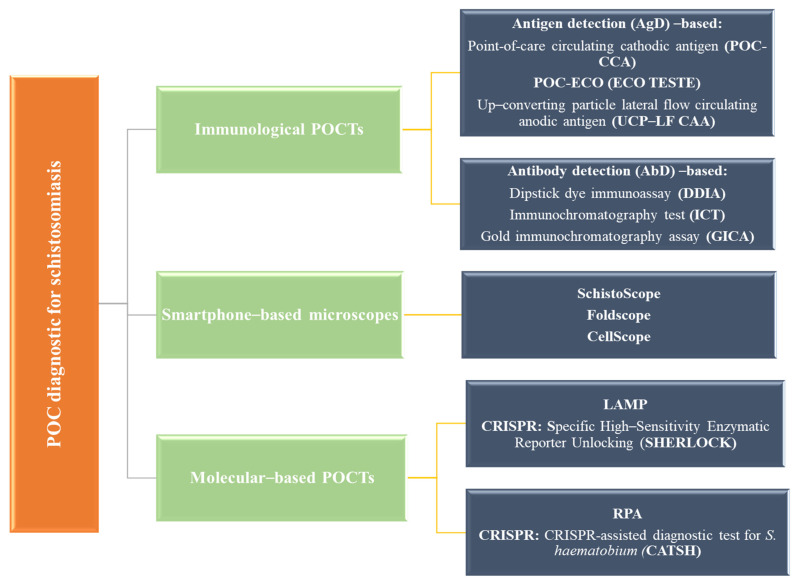
The current and upcoming POC diagnostic assays for schistosomiasis.

**Table 2 tropicalmed-09-00243-t002:** Conventional techniques with their mechanism, advantages, and drawbacks.

Conventional Techniques	Mode of Action	Advantages	Disadvantages	References
Stool and urine examination	Detection of parasite eggs in stool or urine samples using microscopy	Direct evidence of infection Low operational cost and feasible in low-resource settings	Low sensitivity in low-intensity infections Requires multiple samples for increased accuracy	[64,65]
Rectal biopsy	Microscopic examination of rectal tissue for parasite eggs	Higher sensitivity than stool examination in low-intensity infections	Invasive procedure Not practical for large-scale screening	[64,66]
Serological diagnosis: immunoenzymatic assays (ELISA)	Detection of antibodies or antigens in blood samples	High sensitivity and specificity, especially with modified ELISA techniques; useful in low-transmission areas	Cannot distinguish between past and current infections; potential for false positives due to cross-reactivity	[65]
Serological diagnosis: CCA test	Detection of CCA in serum using monoclonal antibodies	High specificity and sensitivity for active infections; useful for monitoring treatment efficacy	Limited availability and higher cost	[67]
Imaging techniques: ultrasound	Imaging of affected organs to assess schistosomiasis-related pathology	Non-invasive and provides real-time results; portable and cost-effective compared to other imaging methods	Requires trained personnel; less effective in detecting early-stage infections	[68,69]
Imaging techniques: computed tomography and magnetic resonance imaging (MRI)	Detailed imaging of internal organs to detect schistosomiasis complications	High accuracy in detecting ectopic forms of the disease; better tissue differentiation with MRI	High cost and limited accessibility; requires specialized equipment and personnel	[68]
Hematuria detection (reagent strips and visual examination)	Detection of blood in urine as an indicator of *S. haematobium* infection	Simple and cost-effective for field use; it can be performed with minimal training	Less specific, as hematuria can result from other conditions; lower sensitivity compared to imaging	[70]

**Table 3 tropicalmed-09-00243-t003:** A summary of the various techniques, their principles, required samples, advantages, and disadvantages, along with their sensitivity and specificity for diagnosing schistosomiasis.

Technique	Principle	Type of Sample Analyzed	Advantages	Disadvantages	Field or Laboratory Use	Sensitivity	Specificity	References
Microscopic (conventional)	Direct observation of parasite eggs	Stool (S), urine (U)	Inexpensive, requires minimal equipment	Time-consuming, low sensitivity for early/minute infections	Laboratory (Lab)	Low (L) (especially in early stages)	High (H) (when parasite eggs are present)	[71]
PCR	Amplifies and detects parasite DNA	Blood (B), U, S,	Highly sensitive, can detect low levels of parasite DNA	Expensive, requires technical expertise	Lab	H	H	[73]
LAMP	Amplifies parasite DNA using isothermal conditions	B, U, S	Rapid, does not require a thermal cycler	Requires trained personnel, potential false positives	Both	H	H	[74]
RPA	Amplifies parasite DNA	B, U, S	Fast, field-deployable, does not require complex instruments	Moderate cost, less validation in field conditions	Both	H	H	[75]
Rapid diagnostic test	Detects antibodies or antigens using immunochromatography	B, U	Simple, rapid, field-deployable	Limited sensitivity and specificity in early infection stages	Field	M	M	
Lateral flow assay	Detects antigens using antibody-labeled particles	B, U	Simple, rapid, portable, field-friendly	Limited sensitivity and specificity	Field	M	M	[76]
Smartphone-based devices	Uses smartphone technology to analyze results from lateral flow assays	B, U	Portable, easy to use, field-deployable	Limited validation and availability	Field	M to H (depends on device)	M	[76]
ELISA	Detects host immune response proteins (antibodies, cytokines, etc.)	B	Can assess host immune response, widely used	Requires laboratory setup, moderate sensitivity	Lab	M	M	[77]
Mass spectroscopy	Detects specific proteins or biomarkers	B, tissue (T)	Highly sensitive, can identify proteins	Expensive, requires complex instruments	Lab	H	H	[78]
Proteomic techniques	Detects schistosome proteins (e.g., SjTs4, MF3, SjPGM, SjRAD23)	B, U	Can differentiate between current and past infections	Expensive, requires high technical expertise	Lab	H	H	[3]
MicroRNA detection	Detects schistosomes-specific microRNAs	B, U, S	High sensitivity	Expensive, requires field validation	Lab	H	H	[72]

Note: Lab—laboratory; H—high; M—moderate; L—low.

**Table 4 tropicalmed-09-00243-t004:** Schistosomal specific DNA and RNA targets and their significance in the molecular detection of human schistosomiasis.

Schistosoma Type	Genetic Target Amplified (DNA/RNA)	Amplification Method	Nature of Sample Utilized for Evaluation	Sensitivity/Specificity	References
*S. japonicum*	Retrotransposon SjR2	Conventional PCR (C-PCR)	Stool Sera/stool		[88]
		Nested-PCR (N-PCR)	Sera		
	Retrotransposon SjR2	LAMP	Sera	Sensitivity: 95.5%	[89]
		RTPCR	Feces		[90]
		RT-PCR	Goat’s Plasma/serum	Sensitivity: 98.74–100%	[91]
	231-bp DNA of retrotransposon SjR2	N-PCR	Animals (goat, buffaloes) sera and dry blood filter paper	Sensitivity: 92–100% Specificity: 97.60%	[92]
	*Retrotransposons SjCHGCS 19* gene	N-PCR	Serum		[93]
	28S rDNA	C-PCR	Urine		[94]
	28S rDNA	LAMP	Snail (DNA)	100 fg	[95]
	28S rDNA		Feces/Urine		[96]
	*Cytochrome oxidase (Cox1)* gene	C-PCR	Sera/urine/saliva		[97]
	Cox2	RT-PCR	Stool		[91]
	Specific regions between NADH dehydrogenase (nad6) and cox2	Multiplex real-time PCR	Stool		[98]
	Specific regions between nad1 and nad2	M-PCR	Stool		[99]
	Nad1	C-PCR RT-PCR	Feces		[100]
	Nad6	RT-PCR	Feces		[101]
	miR-3479, miR-3096, miR-001, miR-277, Bantam	RT-PCR	Plasma/sera		[102]
	miR223	RT-PCR	Serum		[103]
	NADH I (mitochondrial DNA)	RT-PCR	Feces	1 EPG	[104]
	18S rDNA	RT-PCR	Mouse feces and serum	10 fg	[105]
*S. haematobium*	Dra 1 repeats	RT- PCR	Urine	Sensitivity: ~80%	[106]
	Dra I (DQ157698)	PCR		1 ng	[107]
	Cox1	C-PCR	Sera/urine/saliva/semen		[108]
		RT-PCR	Lavage fluid of the vagina		[108]
		M-PCR	Stool		[109]
	Internal transcribed spacer rDNA ITS	C-PCR	Urine		[110]
	ITS2 rDNA region	RT-PCR	Urine		[84]
	NADH-3 (mitochondrial DNA)	PCR	Urine	1 pg	[111]
*S. mansoni*	121-bp tandem repeat sequence	C-PCR	Sera		[108]
			Stool		[112]
			Urine		[112]
		Touch down PCR	Serum		[113]
		RT-PCR	Serum		[114]
			Cerebrospinal fluid		[115]
	28S rDNA	PCR-ELISA	Feces		[108]
	28S rDNA	Multiplex PCR	Mice urine	Sensitive: 94.4% Specificity: 99.9%	[116]
		C-PCR	Urine	10 copies/μL of *S. haematobium*	[117]
	18S rDNA	Nested PCR	(Snail’s DNA)	10 fg	[118]
	18S rDNA	PCR		40 pg/μL Sensitivity: 94.4% Specificity: 99.9%	[119]
	Cox1	M-PCR	Feces		[109]
	Nad1	RT-PCR	Feces		[108]
	NADH dehydrogenase (nad5)	M-PCR	Feces		[108]
	Specific regions between nad6 and cox2	M-PCR	Stool		[120]
	Mitochondrial minisatellite DNA sequence (620 bp)	LAMP	Feces		[84]

**Table 5 tropicalmed-09-00243-t005:** Selected LFIAs for the diagnosis/screening of schistosomiasis.

Antigen/Antibody Detection-Based Test	Test Used	Target and Species	Sensitivity	Specificity	References
AgD	POC-CCA	CCAs *S. mansoni* *S. haematobium* *S. japonicum*	29–99% for different species	35–95%	[124]
AbD soluble egg antigen (SEA)	DDIA	Anti-SEA Abs *S. japonicum*	90.4–95.3	45.9–62%	[125]
AbD (purified extracts)	*Schistosoma* ICT IgG–IgM	Abs against partially purified Ag isolated from crude lysates of *S. mansoni* *S. haematobium* *S. mekongi* *S. intercalatum/guineensis*	94–100	62–63.9	[126]
AgD	Urine CCA (Schisto) Eco Teste^®^ (POC-ECO)	CCAs *S. mansoni*	90.8	87.9	[127]
AbD (crude extracts)	*Sj*-ICT	Anti-AWSE Abs *S. japonicum*	90.8	87.9	[128]
AgD	UCP-LF-CAA	CAA *S. haematobium* *S. mansoni*	80–97%	90–100	[129]
AbD (crude extracts)	*Smk*-ICT	Anti-AWSE Abs *S. mekongi*	78.6	97.6	[130]
AbD (SjSAP4 recombinant protein)	GICA	Abs against SjSAP4 *S. japonicum*	83.3–95	100	[131]
AbD (SEA)	Dipstick with Latex Immunochromatographic Assay (DLIA)	Anti-SEA Abs *S. japonicum*	95.1	94.91	[131]
AbD (purified extracts)	GICA	Abs against partially purified SEA (>10 kDa fragments) *S. japonicum*	93.7	97.6	[132]
AbD (recombinant proteins)	POC-ICTs	Abs against Sh-TSP-2 and MS3_01370 *S. haematobium*	75–89	100	[133]

**Table 6 tropicalmed-09-00243-t006:** Summarized information of the drug evaluation, alone or in combination, along with their effect on the treatment of schistosomiasis.

Treatment	Category	Species Targeted	Efficacy/Details	References
PZQ	Standard drug	Most *Schistosoma* spp.	High efficacy with 40 mg/kg, CR: 57–88%, egg reduction rates: 95%	[138]
PZQ (60 mg/kg)	Standard drug	*S. japonicum*, *S. mekongi*	Higher dosages for these species	[138]
PZQ + antimalarial artemisinin	Drug Combination	Immature larval forms of schistosomes	Higher efficacy and cure rates than PZQ alone	[141,155]
PZQ + Dihydroartemisinin-Piperaquine (DHP)	Drug Combination	Not specified	Superior effectiveness, higher systemic PZQ exposure	[141,143,155]
Linalool or *Cinnamomum camphora* extracts	Snail management (Preventive)	Snail hosts for schistosomes (*S. japonicum*)	Disintegration of snail gills and cell degradation, used for *S. japonicum*	[144]
PZQ + Moxidectin	Drug combination	*S. mansoni* and *S. haematobium*	CR: 92.9% (*S. mansoni*), 100% (*S. haematobium*), >99% egg reduction. PZQ damages the worm’s tegument; moxidectin affects the nervous system	[135]
Artesunate-PZQ	Drug combination	*S. mansoni*	Promising results	[147]
Artesunate-Mefloquine	Drug combination	*S. haematobium*	Effective treatment	[148]
PZQ (intradermal administration)	Alternative delivery	Not specified	Good tolerability, potential for improved adherence	[149]
LAI PZQ formulations	Alternative delivery	Not specified	Potential to simplify treatment regimens, improve adherence	[149]
LAI (PZQ implants)	Alternative delivery	*S. japonicum*	Stable plasma concentrations maintained for up to 70 days, preventing infection in mice	[151]
Single vs. multiple doses of PZQ	Optimized existing regimen	Different *Schistosoma* species	Enhanced outcomes, especially in preschool-aged children	[152]
Next-generation PZQ derivatives	PZQ derivatives	Various *Schistosoma* spp.	Improved efficacy and longer-lasting effects	[154]
Water Sanitation and Hygiene Programs (WASH)	Integrated with Drug Treatment	General schistosomiasis prevention	Timing and integration with sanitation efforts being explored	[153]
PZQ in Cohort Study	Evaluated Effect on LGIM	*S. mansoni*	Partial improvement in LGIM abnormalities 6 months after treatment	[145]
Health education programs	Complementary strategy		Essential for promoting early diagnosis, treatment adherence, and understanding of drug combinations, sanitation, and hygiene	

**Table 8 tropicalmed-09-00243-t008:** Current status of different schistosomal vaccines in the clinical trials.

Vaccine Candidate	Species Targeted	Target Antigen	Clinical Trial Phase	References
r Sm-14/GLA-SE (r = recombinant)	*S. mansoni*	Glutathione S-transferase (GST) from *S. mansoni*	Phases I and IIa are complete. Phase IIb started.	[156]
rSh28GST/Alhydrogel^®^ (Bilharvax)	*S. haematobium*	Glutathione S-transferase (GST) from *S. haematobium*	Phases I, II, and III ended.	[159]
r Sm-p80/GLA-SE	*S. mansoni*	Sm-p80 antigen (large subunit of calpain)	Phase I started. Evaluation for safety and immunogenicity.	[156]
rSm-TSP-2/Alhydrogel^®^	*S. mansoni*	Sm-TSP-2 antigen	Phase Ia finished. Phase Ib started. Safety and efficacy in human subjects are evaluated.	[162]
Multi-epitope peptide-based vaccine	*S. mansoni*	Transmembrane proteins of *S. mansoni*	Preclinical.	[173]

## Data Availability

Not applicable.

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
