# Peer review of "A Recent Advance in the Diagnosis, Treatment, and Vaccine Development for Human Schistosomiasis"

_tropicalmed, 2024, doi:10.3390/tropicalmed9100243_

Round 1

Reviewer 1 Report

Comments and Suggestions for Authors

I have suggested that the authors consider other references as aspects that I think are important for schistosomiasis control are not covered in this review and it could/should be that more comprehensive

Comments on the Quality of English Language

The English is not adequate. If you had sent a Word file I would have corrected the English as I read the manuscript. I started writing out the corrections but it took too long. I've sent what I suggest should be changed in the beginning but the paper needs to be checked.

Author Response

Note:

Reviewer 1: All the changes are highlighted in yellow highlights in the manuscript

Reviewer 1

The review is divided into 10 sections, and the latter three chapters on Diagnosis, Treatment and Vaccines are informative in identifying and explaining the many new technologies that have been developed. However, the problem of schistosomiasis occurs in the developing world as a result of inadequate clean water and sanitation and the reality is that the WHO target dates for the elimination of schistosomiasis have needed to be regularly extended. The authors quote 2020 and 2025 but the 2030 Sustainable Development Goals’ target is also approaching. Although authors of this review need to consider this problem, (which they have noted in the introduction), they have not included the ‘Schistosomiasis Control Initiative’ in their review and this effort to treat millions of children in Africa needs consideration [The Schistosomiasis Control Initiative (SCI): Rationale, development and implementation from 2002-2008. (Fenwick et al, 2012)]. There are also two other perspectives which the authors should consider concerning the Mass Treatment Approach. The Kokaliaris et al. (2022) points to successes from this strategy whereas the Onasanya et al’s paper suggests a different approach. In this schistosomiasis review it is important that the authors of the review present the evidence supporting both positions. I am providing details for the authors to consider.

Kokaliaris C, Garba A, Matuska M, Bronzan RN, Colley DG, Dorkenoo AM, Ekpo UF, Fleming FM, French MD, Kabore A, Mbonigaba JB, Midzi N, Mwinzi PNM, N'Goran EK, Polo MR, Sacko M, Tchuem Tchuenté LA, Tukahebwa EM, Uvon PA, Yang G, Wiesner L, Zhang Y, Utzinger J, Vounatsou P. Effect of preventive chemotherapy with praziquantel on schistosomiasis among school-aged children in sub-Saharan Africa: a spatiotemporal modelling study. Lancet Infect Dis. 2022 Jan;22(1):136-149. doi: 10.1016/S1473-3099(21)00090-6. Epub 2021 Dec 2. Erratum in: Lancet Infect Dis. 2022 Jan;22(1):e1. doi: 10.1016/S1473-3099(21)00771-4.

Onasanya A, Bengtson M, Oladepo O, Van Engelen J, Diehl JC. Rethinking the Top-Down Approach to Schistosomiasis Control and Eliminat ion in Sub-Saharan Africa. Front Public Health. 2021 Feb 18;9:622809. doi: 10.3389/fpubh.2021.622809.

AR: Thank you for this valuable suggestion. The author has taken valuable information from the provided papers and accordingly revised the manuscript.

Abstract: The control and elimination of schistosomiasis have over the last two decades involved several strategies, with the current strategy by the World Health Organization (WHO) focusing mainly on treatment with praziquantel during mass drug administration (MDA). However, the disease context is complex with an interplay of social, economic, political, and cultural factors that may affect achieving the goals of the Neglected Tropical Disease (NTD) 2021-2030 Roadmap. There is a need to revisit the current top-down and reactive approach to schistosomiasis control among sub-Saharan African countries and advocate for a dynamic and diversified approach. This paper highlights the challenges of praziquantel-focused policy for schistosomiasis control and new ways to move from schistosomiasis control to elimination in sub-Saharan Africa. We will also discuss an alternative and diversified approach that consists of a Systems Thinking Framework that embraces intersectoral collaboration fully and includes co-creating locally relevant strategies with affected communities. We propose that achieving the goals for control and elimination of schistosomiasis requires a bottom-up and pro-active approach involving multiple stakeholders. Such a pro-active integrated approach will pave the way for achieving the goals of the NTD 2021-2030 roadmap for schistosomiasis, and ultimately improve the wellbeing of those living in endemic areas.

Although the reader gets the gist of the review, the English requires editing. (It would however, be preferable to send the document in ‘Word’ so that the reviewer could indicate the many suggested corrections more easily.) I noted some of the suggested English and Content changes that are required, and provide these as examples:

Abstract:

Line 34, lack of adequate sanitation (adds clarity)

Line 36, a disease of poverty , not ‘the’ disease as there are many others

Line 38, species not members is the preferred word. There are different Schistosoma species and infections may be single or comprise both S. haematobium and S.mansoni.

Line 42, Female genital schistosomiasis (FGS) has been inadequately addressed in this review, as explained below..

Line 48. Praziquantel is less active against juvenile than mature Schistosoma parasites (This is an important point regarding the need for retreatment) (Vale et al, 2017)

AR: Thank you for this valuable suggestion and comments. The author has corrected all these errors in the revised manuscript as suggested by the respected reviewer.  

  1. Introduction:

Line 71/72. WHO has set global goals for schistosomiasis as a public health problem……

Also lines 75/76, 78, 79, 81,

Line 90. The abbreviation for ‘preventive chemotherapy’ (PCT) needs to be standardized.

Line 99, no ‘a’ concerning swimmers’ itch.

Line 104, In one of the studies it has also been observed….

Lines 128, 130, 134 and 148 need English corrections.

Line 154 should be changed to ‘female and male adult parasites’ as they are separate at this stage of their lifecycle.

Lines 167/168 are unclear. The authors state ‘where they enter the skin of the human host’ which is accurate, but the authors should delete ‘may behave as contagious forms for people’.

Line 171, ‘ for the cercariae to be released’.

Lines 177/178, Not only the urinary tract - FGS is a gynecological disease in women, but should be initially explained as above.

Line 216, ‘In the Asian…’. Line 218, In Asia (delete among)

Line 224, Puerto Rico is NOT a South American country.

Line 268, 95% Confidence Interval is abbreviated to 95% CI not CRI .

Line 671. Point of Care is POC not POCT?

AR: Thank you for this valuable suggestion and comments. The author has all the necessary changes, including English language, in the revised manuscript as suggested by the respected reviewer.  

ISSUES CONCERNING THE REVIEW

GAP IN THE CONTENT: FEMALE GENITAL SCHISTOSOMIASIS

Line 111. The authors discuss the damage caused but the schistosome eggs but they have not adequately covered urogenital schistosomiasis The WHO reviewed and changed the description of urinary schistosomiasis to urogenital schistosomiasis because of the genital problems resulting from schistosomiasis, which can be mistakenly diagnosed as a sexually transmitted infection or cervical cancer (Engels D, et al. Integration of prevention and control measures for female genital schistosomiasis, HIV and cervical cancer. Bull World Health Organ. 2020;98(9):615-624).

  1. SCHISTOSOMA HAEMATOBIUM LIFECYCLE

The schistosome lifecycle does not clearly state that figure 1b provides details ONLY of the lifecycle of Schistosoma mansoni.

An additional figure 1c is required to show the lifecycle of Schistosoma haematobium which migrates mainly to the urinary and genital tracts.

AR: Thank you for this valuable suggestion and comments. The author has modified the life cycle of the Schistosomes and emphasised both S. japonicum and S. hematobium.

Moreover, the authors have also added 2 new figures dedicated to S. japonicum and S. hematobium, as suggested by the respected reviewer.

  • PATHOPHYSIOLOGY
  • The pathophysiology caused by urogenital schistosomiasis (UGS) is inadequately reviewed. Authors to please check the following two references:

Kjetland EF, Leutscher PD, Ndhlovu PD. Trends Parasitol., 28 (2012). 58-65

Orish N, Komla Senanu Morhe E, Azanu W, Alhassan RK, Gyapong M. The parasitology of female genital schistosomiasis. Current Research in Parasitology and Vector-borne Diseases, 2 (2022), 100093

AR: Thank you for this valuable suggestion and comments. The author has now explained urogenital schistosomiasis (UGS) with relevant references in the revised manuscript as suggested by the reviewer.

  • Authors to please note that FGS is NOT cured by drugs and needs to be prevented.

AR: Thank you for pointing out this mistake. The authors have noted and also mentioned the same in the revised manuscript, as suggested by the respected reviewer.  

  1. DISTRIBUTION: AUTHORS TO PLEASE RECONSIDER

Line 227. It is inaccurate to say that S.mansoni is the main specie in Sub-Saharan Africa as it depends on which country because schistosomiasis is very focal.

Line 235, In Sub-Saharan Africa (SSA) both Schistosoma haematobium and Schistosoma mansoni can be prevalent simultaneously.

Line 299-303. FGS is inadequately covered in this section.

Lines 331-348. It would be useful if the authors included references for this section.,

Lines 339-340. It is unclear what the authors ean – successful treatment of which disease or both?

Line 341, It is essential that there is a differential diagnosis concerning FGS, STIs and cervical cancer

Lines 404/405, The prevalence of Schistosoma haematobium and Schistosoma mansoni is unclear

AR: Thank you for this valuable suggestion and comments. The author has incorporated all the above comments in the revised am NoScript as suggested by the respected reviewer.

Authors have now added a complete paragraph related to FGS in this section as suggested by the respected reviewer.

“The prevalence of Schistosoma haematobium and Schistosoma mansoni is unclear” Now, the authors have rephrased the sentences for better understanding.

  1. SIGNS AND SYMPTOMS

The unanswered question from this review is whether the population affected by schistosomiasis is able to link the signs (blood in urine) and symptoms (difficulty in urinaton) of the disease (which is well-known in the vernacular) in most countries, but few link the signs and symptoms with the snails in the water that they frequent. The review makes no reference to the health education that should accompany the diagnosis, treatment and hopefully, vaccination.

AR: Thank you for this valuable comment and suggestion. The authors have added the role of health education in the prevention, diagnosis, treatment and use of vaccines for schistosomiasis in t all the sections of the revised manuscript, as suggested by the respected reviewer.

  1. MORBIDITIES AND CO-MORBIDITIES

The problem of the morbidity/co-morbidity is the stigma associated in many SSA countries both with HIV and in some societies, because of the gynecological symptoms similar to STIs, with schistosomiasis (Lambert al., 2024, Frontiers in Tropical Diseases)

AR: Thank you for this suggestion. The authors have now modified the section with the valuable information in the revised manuscript as suggested by the reviewer.

  1. RECENT OUTBREAKS

  1. DIAGNOSIS

The new methods are promising and important because women with FGS may not excrete schistosome eggs.

AR: The authors have added 1 new table in the treatment section in the revised manuscript, as suggested by the respected reviewer. Moreover, the authors have also updated the FGS section in the revised manuscript.

  1. TREATMENT

It is interesting that despite the fears of resistance to praziquantel over many decades, this has yet to be reported.

AR: The authors have added 1 new table in the treatment section in the revised manuscript, as suggested by the respected reviewer.

  1. VACCINES

It is very important that communities understand how vaccines work and the benefits to ensure uptake when eventually they become available. The backlash against the COVID-19 vaccine emphasizes the importance of promoting new technologies and developing te population’s confidence in their use.

AR: Thank you for this valuable comment and suggestion. The authors have now added recent studies in all of these sections with new figures and tables.

 added the role of health education in the prevention, diagnosis, treatment and use of vaccines for schistosomiasis in t all the sections of the revised manuscript, as suggested by the respected reviewer.

The authors have updated the vaccine status and treatment section as per the recent study.

Reviewer 2 Report

Comments and Suggestions for Authors

The manuscript "A Recent advance in the diagnosis, treatment, and vac-2 cine development for human schistosomiasis" extendly review the features related to infection caused by schistosomes. More or less of half of the manuscript is dedicated to the description of parasites (life cycle, etc), geographical distribution and dedicates a lot of part to describe the pathology, etc, etc. Consider the title of the manuscript the topics that should be emphasized are diagnostic, treatment and vaccines.

At this point the manuscript is a little bit long and I think that you can shorter by resume the first 7 sections. Also, I think that it is important to re-organize the sections (see comments below) in order to render the manuscript more readable and lesss confuse. In some sections you describe the disease and two sections ahead you have a similar theme. I do believe that you can also improve your figures. In general, I think that the manuscript could be improved.

Line 38 – Rephrase the sentence “The major infections are caused …either alone or together with different species”. Consider using “…Three Schistosoma species (S. mansoni, S. japonicum and S. haematobium) cause significant human infections. Co-infections with Schistosoma and other parasites are widely common.”

Line 53 – Replace diagnosis of “…schistosomal illness…” by schistosomiasis.

Line 55 -  Replace “…the vaccine…” with “…the vaccine development.”

Line 60 – Replace “…originates through…” by “…caused by trematode flatworms…”.

Line 61 – Replace “….illness that leads to chronic ill health” with “…Katayama fever, a poverty-based illness that if untreated leads to life-threatening pathologies”.

Line 74 – Replace schistosomal infections with schistosomiasis.

Line 75 – Replace with  “..are endemic in 78 countries of which 51 have moderate-to-high transmission that requires preventive treatment.”

Line 76 – Remove “…this report..” and begin the sentence with “Moreover, also claims..”.

Line 77 – Replace “…impacted…” with “…affect…”.

Line 82 – Please consider “ Comparing data from 2019 and 2020 (COVID-2019 pandemic), during COVID-19 pandemic corresponding a 27% decrease in treatment coverage…”.

Line 88 – Replace “…schistosomiasis diseases…” by”…schistosomiasis…”. It is repetitive to say schistosomiasis diseases since schistosomiasis is an infection caused by Schistosoma. Please, correct this through the manuscript.

Line 98 – Similarly,  do not use schistosomal parasite. You can use only parasite. Also, it affects not only swimmers but, as you mention in  the abstract, persons who use infected lake water to wash clothes, obtain water, etc.

Line 101-102 -  This only occurs in case of infection with S. japonicum?

Line 103-105 – You should add the information that schistosomiasis japonica is nearly to eradication in China. What is the relevance of this sentence? I don’t think that is necessary here. Consider either clarifying and stating the importance of this information or removing it.

Line 107 – Replace “…the whole word…” with “…worldwide…”. Consider removing the sentence “For instance, Song et al., emphasized….”. You can complete the previous sentence with this information and reference.

Line 110 -  Rephrase this sentence. For example, “The inflammatory responses to the presence of eggs produced by adult worms are responsible for infection-associated pathologies.”

Line 116 – Clarify this sentence. Do you pretend to say that despite schistosomiaisis,  anemia and malnutrition are frequent conditions in endemic locations?

Line 119 – Remove “…a team led by Gruninger…”. You can replace it with “A cross-sectional investigation demonstrated a high prevalence of schistosomiasis in…”.

Line 144   Replace “…around the globe...” with worldwide.

Line 149 Correct through the manuscript life cycle and not lifecycle.

Line 154 – Replace “These larval forms enter the body…” with “…enter into definitive host...”.

Line 159 – Could you improve this figure? You can merge the two images otherwise they are a little bit repetitive.

Line 163 –  Replace “…that infect the infectious agent (snail)” with “..that infect the intermediate host (snail).”.

Line 164  There is a gap between the release of the cercariae until schistosomula. You should enter that when cercariae contacts with mammal skin, release their tale and the head (schistosomula) penetrate into skin. Please, rewrite lines from 165 until 171. It is a little bit messy and confusing. You mentioned cercariae, then schistosomula, then back to the asexual part in the snails. It must follow logic and not go back and forth in the life cycle.

Line 173 – You should be clear that most of the species of  Schistosoma cause intestinal schistosomiasis (ITS) and S. haematobium is associated with urogenital schistosomiasis. I suggest that when you describe the parasite life cycle mention that most of the species mature on mesenteric veins and S. haematobium on the vesical plexus of the bladder. Otherwise, this sentence gives the wrong idea that the 5 species of Schistosoma could cause ITS or UGS. Or, since you mention this information ahead, you can consider this “Schistosomiasis can be categorized as intestinal schistosomiasis (ITS) and UGS. ITS is characterized by intestinal damage, hypertension in the abdominal blood vessels, liver enlargement, among others. UGS mainly affects….”. It is important to mention that Sh-infection is associated with the development of bladder cancer.

Line 178-180 What do you want to say with this sentence? It is a little bit confusing.

Line 182 – Please remove “…and from there…”. Replace with “…larva enters the skin and migrates…”.

Line 189 and 190 – Remove “...then...”, and replace “…pierce…” with “…transverse…”.

Line 195 Rephrase this sentence. Consider this: “The formation of a granuloma that consists in an agglomeration of white blood cells (macrophages, lymphocytes, eosinophils and fibroblasts) that surround and isolate the eggs and releases cytokines…matrix proteins. The cytokines and chemokines are responsible for the modulation…, while collagen and matrix proteins are associated with fibrosis…organ. Together, these features are responsible for…the organ”.

Figure 2 – Can you improve this figure? Lower the letter size, and replace “beta” with the respective symbol.

Line 216 - 225 I think you can remove this paragraph. It becomes repetitive because you describe and mention the countries and continents below. Another option is to merge this information into the paragraph of line 226.

Line 240 – Correct schistosomal parasite.

Line 262 – Remove 2015. It should be “Lai et al. [25],..”

Line 274 – I think that this section 5 should be before pathophysiology. I think you can merge sections 5 and 3. Otherwise, it is somehow repetitive.

Lines 342-348 You should include references that support your statements.

Line 397 Correct schistosomal illness and replace it with schistosomiasis.

Line 400 Please, define SAC. And define the years of the outbreaks.

Line 438-449 You should be more descriptive about the tests you mentioned, such as how they work and their advantages and disadvantages. Same for the subsections 8.2 and 8.3.

Line 508 – Remove the parenthesis from SjTs4 and MF3.

Line 544 – Remove 2005 or insert the correspondent reference number.

Line 555 –  Correct schistosomal sps to Schistosoma spp or parasites.

You should consider including a table that resumes all the diagnostic techniques described in the manuscript. The table must include technique, technique principle, type of sample that is analyzed, advantages and disadvantages, used in the field or laboratory, sensitivity and specificity, similar to Table 2.

Line 596 – Add a final stop between “S. japonicum” and “Additionally”.

Table 2 – Improve the table, for example, amplification, the n letter is separated.

Line 718 – What do you want to mean with “One major drawback of this drug is that its cercarial exposure time is unknown”? Please clarify this.

Section 9 – Please consider inserting a table that resumes all the treatments evaluated and described here and specifies if it is a drug combination, drug repurposed, PZQ derivatives, etc.

Lines 753, 754, 756 Is 76.2%? Why a point is between the numbers?

Sections 10 and 11 – I think you shouldn’t separate these sections. It is more suitable if you merge it.

Line 832 – The sentence “To date…animal models.” should appear in the same paragraph as “Several investigations…40-60%”.

Line 848 Do you want to mean 100 lg? What is the significance of lg?

Line 849 Instead “A group led by Mo…” use “Clinical trials assessed the…”.

Line 931 – Replace 50 ug by μg.

Table 4 – Why doesn’t proteomics have any reference? Same for immunomics.

Line 972 – Correct the word S. mansoni.

Line 988 – I think you should include a brief discussion of why there are more vaccine candidates for S. mansoni then for S. haematobium. What are the difficulties through vaccine development and our point of view about the direction that should be pursue. The section 14 should appear here.

Section 13 – I am not sure if this section should appear here. I believe that it could be just before the description of treatment and treatment alternatives as vaccines.

Line 1046 – I think the sentence “Molecular methods..” should appear after “Moreover, environmental….outbreak”. Also you should link the subjects.

Comments on the Quality of English Language

Moderate editing of English language required.

Author Response

Note:

Reviewer 2: All the changes are highlighted in cyan color highlights

Reviewer 2

The manuscript "A Recent advance in the diagnosis, treatment, and vaccine development for human schistosomiasis" extendly review the features related to infection caused by schistosomes. More or less of half of the manuscript is dedicated to the description of parasites (life cycle, etc), geographical distribution and dedicates a lot of part to describe the pathology, etc, etc. Consider the title of the manuscript the topics that should be emphasized are diagnostic, treatment and vaccines.

At this point the manuscript is a little bit long and I think that you can shorter by resume the first 7 sections. Also, I think that it is important to re-organize the sections (see comments below) in order to render the manuscript more readable and lesss confuse. In some sections you describe the disease and two sections ahead you have a similar theme. I do believe that you can also improve your figures. In general, I think that the manuscript could be improved.

Line 38 – Rephrase the sentence “The major infections are caused …either alone or together with different species”. Consider using “…Three Schistosoma species (S. mansoni, S. japonicum and S. haematobium) cause significant human infections. Co-infections with Schistosoma and other parasites are widely common.”

AR: Thank you for this valuable comment and suggestion. The authors have now rephrased the suggested sentences in the revised manuscript as suggested by the reviewer.

Line 53 – Replace diagnosis of “…schistosomal illness…” by schistosomiasis.

Line 55 -  Replace “…the vaccine…” with “…the vaccine development.”

Line 60 – Replace “…originates through…” by “…caused by trematode flatworms…”.

Line 61 – Replace “….illness that leads to chronic ill health” with “…Katayama fever, a poverty-based illness that if untreated leads to life-threatening pathologies”.

Line 74 – Replace schistosomal infections with schistosomiasis.

Line 75 – Replace with  “..are endemic in 78 countries of which 51 have moderate-to-high transmission that requires preventive treatment.”

Line 76 – Remove “…this report..” and begin the sentence with “Moreover, also claims..”.

Line 77 – Replace “…impacted…” with “…affect…”.

AR: Thank you for this valuable comment and suggestion. The authors have replaced the words with suggested words in the sentences in the revised manuscript, as suggested by the reviewer.

Line 82 – Please consider “ Comparing data from 2019 and 2020 (COVID-2019 pandemic), during COVID-19 pandemic corresponding a 27% decrease in treatment coverage…”.

AR: Thank you for this valuable comment and suggestion. The authors have now modified the sentences as per the suggestion of the respected reviewer in the revised manuscript.

Line 88 – Replace “…schistosomiasis diseases…” by”…schistosomiasis…”. It is repetitive to say schistosomiasis diseases since schistosomiasis is an infection caused by Schistosoma. Please, correct this through the manuscript.

AR: Thank you for pointing out this mistake. The authors have now replaced the “…schistosomiasis diseases…” with”…schistosomiasis wherever possible in the revised manuscript as suggested by the respected reviewer.

Line 98 – Similarly,  do not use schistosomal parasite. You can use only parasite. Also, it affects not only swimmers but, as you mention in  the abstract, persons who use infected lake water to wash clothes, obtain water, etc.

AR: Thank you for this valuable comment and suggestion. The authors have now replaced the “…schistosomiasis parasite…” with”…parasite in the whole manuscript as suggested by the respected reviewer.

Line 101-102 -  This only occurs in case of infection with S. japonicum?

AR: Thank you for pointing out this mistake. The authors have now rectified the mistake in the revised manuscript as suggested by the reviewer.

Line 103-105 – You should add the information that schistosomiasis japonica is nearly to eradication in China. What is the relevance of this sentence? I don’t think that is necessary here. Consider either clarifying and stating the importance of this information or removing it.

AR: Thank you for this valuable comment and suggestion. The authors have now modified/removed these sentences in the revised manuscript as suggested by the respected reviewer.

Line 107 – Replace “…the whole word…” with “…worldwide…”. Consider removing the sentence “For instance, Song et al., emphasized….”. You can complete the previous sentence with this information and reference.

AR: Thank you for this valuable comment and suggestion. The authors have now replaced the “whole world” word with “worldwide” in the revised manuscript, as suggested by the respected reviewer.

Line 110 -  Rephrase this sentence. For example, “The inflammatory responses to the presence of eggs produced by adult worms are responsible for infection-associated pathologies.”

AR: Thank you for this valuable comment and suggestion. The authors have now rephrased the said sentences in the revised manuscript, as suggested by the respected reviewer.

Line 116 – Clarify this sentence. Do you pretend to say that despite schistosomiaisis,  anemia and malnutrition are frequent conditions in endemic locations?

AR: Thank you for this valuable comment and suggestion. The authors have simplified the sentences for better understanding in the revised manuscript, as suggested by the respected reviewer.

Line 119 – Remove “…a team led by Gruninger…”. You can replace it with “A cross-sectional investigation demonstrated a high prevalence of schistosomiasis in…”.

AR: Thank you for this valuable suggestion. The authors have now made the necessary changes in the said sentences in the revised manuscript.

Line 144   Replace “…around the globe...” with worldwide.

AR: Thank you for this valuable comment and suggestion. The authors have now replaced the “around the globe” word with “worldwide” in the revised manuscript, as suggested by the respected reviewer.

Line 149  Correct through the manuscript life cycle and not lifecycle.

AR: Thank you for pointing out this mistake. The authors have now rectified all the errors related to this in the revised manuscript, as suggested by the respected reviewer.

Line 154 – Replace “These larval forms enter the body…” with “…enter into definitive host...”.

AR: Thank you for this valuable comment and suggestion. The authors have made the suggested changes in the revised manuscript.

Line 159 – Could you improve this figure? You can merge the two images otherwise they are a little bit repetitive.

AR: Thank you for this valuable suggestion and comments. The author has added 2 new figures dedicated to S. japonicum and S. hematobium, as suggested by the respected reviewer in the revised manuscript.

Line 163 –  Replace “…that infect the infectious agent (snail)” with “..that infect the intermediate host (snail).”.

AR: Thank you for this valuable suggestion and comments. The authors have now made the suggested changes in the revised manuscript.

Line 164  – There is a gap between the release of the cercariae until schistosomula. You should enter that when cercariae contacts with mammal skin, release their tale and the head (schistosomula) penetrate into skin. Please, rewrite lines from 165 until 171. It is a little bit messy and confusing. You mentioned cercariae, then schistosomula, then back to the asexual part in the snails. It must follow logic and not go back and forth in the life cycle.

AR: Thank you for this valuable suggestion and comments. The authors have now thoroughly edited this section in the revised manuscript suggested by the respected reviewer.

Line 173 – You should be clear that most of the species of  Schistosoma cause intestinal schistosomiasis (ITS) and S. haematobium is associated with urogenital schistosomiasis. I suggest that when you describe the parasite life cycle mention that most of the species mature on mesenteric veins and S. haematobium on the vesical plexus of the bladder. Otherwise, this sentence gives the wrong idea that the 5 species of Schistosoma could cause ITS or UGS. Or, since you mention this information ahead, you can consider this “Schistosomiasis can be categorized as intestinal schistosomiasis (ITS) and UGS. ITS is characterized by intestinal damage, hypertension in the abdominal blood vessels, liver enlargement, among others. UGS mainly affects….”. It is important to mention that Sh-infection is associated with the development of bladder cancer.

AR: Thank you for this valuable suggestion and comments. The authors have now thoroughly edited this section in the revised manuscript suggested by the respected reviewer.

Line 178-180 – What do you want to say with this sentence? It is a little bit confusing.

AR: Thank you for this valuable suggestion and comments. The authors have now modified the sentences in the revised manuscript suggested by the respected reviewer.

Line 182 – Please remove “…and from there…”. Replace with “…larva enters the skin and migrates…”.

AR: Thank you for this valuable suggestion and comments. The authors have made the necessary changes in the said sentences in the revised manuscript suggested by the respected reviewer.

Line 189 and 190 – Remove “...then...”, and replace “…pierce…” with “…transverse…”.

AR: Thank you for this valuable suggestion and comments. The authors have made the necessary changes in the said sentences in the revised manuscript suggested by the respected reviewer.

Line 195 – Rephrase this sentence. Consider this: “The formation of a granuloma that consists in an agglomeration of white blood cells (macrophages, lymphocytes, eosinophils and fibroblasts) that surround and isolate the eggs and releases cytokines…matrix proteins. The cytokines and chemokines are responsible for the modulation…, while collagen and matrix proteins are associated with fibrosis…organ. Together, these features are responsible for…the organ”.

AR: Thank you for this valuable suggestion and comments. The authors have rephrased the said sentences in the revised manuscript suggested by the respected reviewer.

Figure 2 – Can you improve this figure? Lower the letter size, and replace “beta” with the respective symbol.

AR: Thank you for this valuable suggestion and comments. The authors have now improved the Figure with the valuable suggestion of the respected reviewer.

Line 216 - 225 – I think you can remove this paragraph. It becomes repetitive because you describe and mention the countries and continents below. Another option is to merge this information into the paragraph of line 226.

AR: Thank you for this valuable suggestion and comments. The authors have now removed the suggested paragraph from the revised manuscript as suggested by the reviewer.

Line 240 – Correct schistosomal parasite.

AR: Thank you for this valuable suggestion and comments. The authors have rectified the errors in the revised manuscript suggested by the respected reviewer.

Line 262 – Remove 2015. It should be “Lai et al. [25],..”

AR: Thank you for this valuable suggestion and comments. The authors have made the necessary suggestions in the revised manuscript suggested by the respected reviewer.

Line 274 – I think that this section 5 should be before pathophysiology. I think you can merge sections 5 and 3. Otherwise, it is somehow repetitive.

AR: Thank you for this valuable suggestion and comments. The authors now shifted and merged this section with 3 in the revised manuscript as suggested by the reviewer.

Lines 342-348  You should include references that support your statements.

AR: Thank you for this valuable suggestion and comments. The authors now provided recent references for the said sentences in the revised manuscript as suggested by the reviewer.

Line 397 – Correct schistosomal illness and replace it with schistosomiasis.

AR: Thank you for pointing out this mistake. The authors now corrected the mistake in the whole manuscript as suggested by the respected reviewer.

Line 400  Please, define SAC. And define the years of the outbreaks.

AR: Thank you for these valuable comments. Here, SAC stands for school-aged children, which is already included in the manuscript. It has been done as it was used several times, and it increased the similarity of the manuscript.

Line 438-449  You should be more descriptive about the tests you mentioned, such as how they work and their advantages and disadvantages. Same for the subsections 8.2 and 8.3.

AR: Thank you for this valuable suggestion and comments. The authors now provided more descriptions of all the possible diagnosis tests in the revised manuscript. Moreover, the authors have also added new tables with their brief descriptions in the revised manuscript as, suggested by the reviewer.

Line 508 – Remove the parenthesis from SjTs4 and MF3.

AR: Thank you for pointing out this mistake. The authors have now corrected the mistake in the revised manuscript.

Line 544 – Remove 2005 or insert the correspondent reference number.

AR: Thank you for pointing out this mistake. The authors have now made the necessary changes in the revised manuscript as suggested by the reviewer.

Line 555 –  Correct schistosomal sps to Schistosoma spp or parasites.

AR: Thank you for pointing out this mistake. The authors have now made the necessary changes in the revised manuscript as suggested by the reviewer

You should consider including a table that resumes all the diagnostic techniques described in the manuscript. The table must include technique, technique principle, type of sample that is analyzed, advantages and disadvantages, used in the field or laboratory, sensitivity and specificity, similar to Table 2.

AR: Thank you for this valuable comment and suggestion. The authors have now added the required tables in both sections as suggested by the reviewer. Moreover, the authors have also modified the table related to the vaccines.  the necessary changes in the revised manuscript, as suggested by the reviewer

Line 596 – Add a final stop between “S. japonicum” and “Additionally”.

AR: Thank you for pointing out this mistake. The authors have now corrected the mistake.

Table 2 – Improve the table, for example, amplification, the n letter is separated.

AR: Thank you for your valuable comment and suggestion. The authors have now improved the table as per the suggestion of the respected reviewer.

Line 718 – What do you want to mean with “One major drawback of this drug is that its cercarial exposure time is unknown”? Please clarify this.

AR: Thank you for your valuable comment and suggestion. The authors have now simplified the said sentences in the revised manuscript as per the suggestion of the respected reviewer.

Section 9 – Please consider inserting a table that resumes all the treatments evaluated and described here and specifies if it is a drug combination, drug repurposed, PZQ derivatives, etc.

AR: Thank you for this valuable comment and suggestion. The authors have now added the required tables in both sections as suggested by the reviewer.

Lines 753, 754, 756 – Is 76.2%? Why a point is between the numbers?

AR: Thank you for pointing out this mistake. The authors have now corrected the mistake in the revised manuscript. As the values were just copied from the previous manuscript and due to the differences in the format and font, this happened.

Sections 10 and 11 – I think you shouldn’t separate these sections. It is more suitable if you merge it.

AR: Thank you for this valuable suggestion. The authors have now clubbed these two sections in the revised manuscript as suggested by the respected reviewer.

Line 832 – The sentence “To date…animal models.” should appear in the same paragraph as “Several investigations…40-60%”.

AR: Thank you for pointing out this mistake. The authors have now corrected this mistake in the revised manuscript as suggested by the reviewer.

Line 848 – Do you want to mean 100 lg? What is the significance of lg?

AR: Thank you for pointing out this mistake. It was actually µg. The authors have now corrected this mistake in the revised manuscript, as suggested by the reviewer.

Line 849  Instead “A group led by Mo…” use “Clinical trials assessed the…”.

AR: Thank you for this valuable suggestion. The authors have now modified the suggestion related to the above sentence in the revised manuscript.

Line 931 – Replace 50 ug by μg.

AR: Thank you for pointing out this mistake. The corrections have been made in the revised manuscript, as suggested by the reviewer.

Table 4 – Why doesn’t proteomics have any reference? Same for immunomics.

AR: Thank you for this valuable suggestion. The authors have now provided the references for both suggestions in the revised manuscript.

Line 972 – Correct the word S. mansoni.

AR: Thank you for pointing out this mistake. Correction has been made in the revised manuscript, as suggested by the reviewer.

Line 988 – I think you should include a brief discussion of why there are more vaccine candidates for S. mansoni then for S. haematobium. What are the difficulties through vaccine development and our point of view about the direction that should be pursue. The section 14 should appear here.

AR: Thank you for this valuable suggestion. The authors have now added a brief discussion about why there are more vaccine candidates for S. mansoni than for S. haematobium. The author has also added the difficulties being faced by the investigators in manufacturing the vaccines related to Schistosomes. The authors have also shifted this section along with the said section in the revised manuscript.

Section 13 – I am not sure if this section should appear here. I believe that it could be just before the description of treatment and treatment alternatives as vaccines.

AR: Thank you for this valuable suggestion. The authors have now shifted this section to the suggested section as per the suggestion of the reviewer.

Line 1046 – I think the sentence “Molecular methods..” should appear after “Moreover, environmental….outbreak”. Also you should link the subjects.

AR: Thank you for this valuable suggestion. The authors have now added it made the necessary changes in the revised manuscript as suggested by the reviewer.

Round 2

Reviewer 1 Report

Comments and Suggestions for Authors

The authors have included the suggested corrections/additions. There are a few additional points. 

Line 99/100 - not only lakes but also river water.

The English really needs careful editing.

The authors need to have obtained permission to adapt figures 1a, 4a and 5. 

Comments on the Quality of English Language

The English really needs careful editing. Maybe different sections were written by different authors as sometimes the English is acceptable whereas at other times the text hardly makes sense. 

It is a long manuscript at 55 pages and editing this will take time. The authors need to address this problem as it distracts from a very interesting and informative review.

Author Response

Reviewer 1

The authors have included the suggested corrections/additions. There are a few additional points. 

  1. Line 99/100 - not only lakes but also river water.

A/R: thank you for pointing out this mistake. The authors have now rectified the mistake in the revised manuscript, as suggested by the respected reviewer.

  1. The English really needs careful editing.

A/R: The authors have now thoroughly edited the manuscript in terms of English editing.

  1. The authors need to have obtained permission to adapt figures 1a, 4a and 5. 

A/R: Thank you for this valuable suggestion.

The figure is completely drawn by authors, where Fig.1 is drawn by using Bio render while Fig.1b and 1c are drawn by using coral draw.

Figure 4a is taken from the open source, MDPI, still authors have now taken permission.

License Number: 5887131166730

The authors have now taken permission for Figure 5 as suggested by the respected reviewer.

Comments on the Quality of English Language

The English really needs careful editing. Maybe different sections were written by different authors as sometimes the English is acceptable whereas at other times the text hardly makes sense. 

It is a long manuscript at 55 pages and editing this will take time. The authors need to address this problem as it distracts from a very interesting and informative review.

A/R: The authors have now thoroughly edited the manuscript in terms of English editing

Reviewer 2 Report

Comments and Suggestions for Authors

Dear Authors,

The manuscript was improved accordingly to reviewer's comments. Therefore, I will suggest (if possible) if there is any possibiity to improve the quality of the figures.

Kinda regards

Comments on the Quality of English Language

Additionally, a moderate review of the english is required.

Author Response

The manuscript was improved accordingly to reviewer's comments. Therefore, I will suggest (if possible) if there is any possibiity to improve the quality of the figures.

 A/R: Thank you for this valuable suggestion. The authors have already improved the quality of all the figures in the revised manuscript, as suggested by the respected reviewer.

The authors have improved the quality of Figure 6 in the revised manuscript.

Comments on the Quality of English Language

Additionally, a moderate review of the English is required.

A/R: The authors have now thoroughly edited the manuscript in terms of English editing
